# LAPLACE SAMPLE INFORMATION: DATA INFORMATIVENESS THROUGH A BAYESIAN LENS

**Johannes Kaiser & Kristian Schwethelm & Daniel Rückert & Georgios Kaissis**
AI in Healthcare and Medicine; Munich Center for Machine Learning (MCML)
Technical University of Munich, Germany
`{johannes.kaiser,g.kaissis}@tum.de`

## ABSTRACT

Accurately estimating the informativeness of individual samples in a dataset is an important objective in deep learning, as it can guide sample selection, which can improve model efficiency and accuracy by removing redundant or potentially harmful samples. We propose *Laplace Sample Information* (LSI) measure of sample informativeness grounded in information theory widely applicable across model architectures and learning settings. LSI leverages a Bayesian approximation to the weight posterior and the KL divergence to measure the change in the parameter distribution induced by a sample of interest from the dataset. We experimentally show that LSI is effective in ordering the data with respect to typicality, detecting mislabeled samples, measuring class-wise informativeness, and assessing dataset difficulty. We demonstrate these capabilities of LSI on image and text data in supervised and unsupervised settings. Moreover, we show that LSI can be computed efficiently through probes and transfers well to the training of large models.

## 1 INTRODUCTION

Deep learning (DL) relies fundamentally on pattern extraction and knowledge acquisition from underlying data to determine a model's parameters. Throughout the training of a neural network, each sample in the training dataset imparts information on these parameters. However, the *amount* of information varies between samples. Beyond providing insights on learning dynamics, measuring per-sample information content allows selecting the most beneficial subsets of a given dataset for model training (i.e. dataset condensation or distillation (Chen et al., 2022; Tan et al., 2023)). This can help to reduce the amount of required computational resources, rendering model training more economical and energy-efficient (Strubell et al., 2019). However, achieving these efficiency goals *without decreasing the learnt model's accuracy* (or other beneficial properties such as sub-group fairness (Ghosh et al., 2023)) is of particular importance and represents a challenge which –so far– has not been sufficiently addressed (Asi & Duchi, 2019). Additionally, evaluating the contribution of individual samples to the final model is of importance across many domains of machine learning (ML) research such as privacy preservation (Ünsal & Önen, 2023), memorization analysis (Bansal et al., 2022), sample difficulty evaluation (Harutyunyan et al., 2021), machine unlearning (Xu et al., 2023), and equitable reimbursement for data contributors (Cummings et al., 2015). All of the aforementioned workflows thus stand to gain from a well-defined notion of sample informativeness.

We contend that a useful measure of sample informativeness should (1) be applicable to most datasets and learning settings without imposing simplifications on the model or constraints on the training and (2) be grounded in information theory to truly measure information and not a related –but different– quantity like sample difficulty. While some previous methods (Agarwal et al., 2022; Jiang et al., 2020) are applicable to a broad range of training settings, they do not measure information flow but rather related quantities (e.g. sample difficulty) which merely *approximate* informativeness. To compute an information-theoretically grounded measure of sample informativeness, the Kullback Leibler (KL) divergence can be employed to approximate the information flow (more formally, the conditional point-wise mutual information) from a single sample to the parameters of the network. However, the KL divergence is defined on probability distributions, whereas most trained neural networks (excluding, e.g., Bayesian neural networks) comprise a single set of parameters, for which the KL divergence is undefined. To enable computing the KL divergence in this setting, prior works have used

techniques like linearization (Harutyunyan et al., 2021). However, this approach leads to performance degradation (Chizat et al., 2019) and scales poorly to larger real-world datasets. Alternatively, noisy iterative learning algorithms, e.g. Langevin dynamics (Negrea et al., 2019) or Gaussian processes (Ye et al., 2023), result in a distribution of parameters but come with a computational cost that increases proportionally with dataset size (Brosse et al., 2018).

To address the aforementioned challenge, we propose *Laplace Sample Information* (LSI) (Definition 2) as a measure of the sample informativeness. LSI estimates the KL divergence-based information via a *Laplace approximation*, a post-hoc method to construct a quasi-Bayesian learner from a trained neural network, and to thus obtain a distribution over model parameters. LSI is applicable to various model architectures, learning settings, and can be computed through automatic differentiation, which facilitates the use of LSI to analyze sample informativeness in diverse DL tasks.

Similar to the aforementioned KL divergence-based techniques, LSI relies on leave-one-out (LOO) retraining of the model to provide a direct measure of informativeness from omitting a single sample. This requires repetitive training, which can become exceedingly costly for larger models. To mitigate this cost, we show that LSI can be computed using a (very) small model as *probe*. The sample order obtained from this probe generalizes well to the training of larger models. This approach allows us to combine the fidelity of a *true* LOO-based informativeness metric while remaining economical and time-efficient. We provide the code as well as the pre-computed values of LSI for common datasets under github.com/TUM-AIMED/LSI. Our contributions can be summarized as follows:

- Our primary contribution is the introduction of LSI, an approximation of the unique information contributed by an individual sample to the parameters of a neural network computed via the KL-divergence of Laplace approximations to the posterior of a neural network's parameters;
- We demonstrate the ability of LSI to identify a spectrum of typical/atypical samples, detect mislabeling, and assess informativeness on the level of individual samples, dataset classes and the entire dataset in supervised and unsupervised tasks across various data modalities (images and text);
- We show that LSI can be efficiently computed by probing the model features and generalizes effectively to large architectures.

## 2 RELATED WORK

**Sample Informativeness** The notion of sample informativeness is defined somewhat inconsistently in literature. For instance, some studies focus on the information contributed by individual samples to a learner, defining informativeness as the reduction in parameter uncertainty upon the addition of a sample (Dwork et al., 2015; Rogers et al., 2016). A more recent body of work describes sample informativeness by extending Shannon information theory to consider computational constraints (Xu et al., 2020; Ethayarajh et al., 2022). Yet another research direction employs influence functions to estimate the informativeness of individual samples. This method examines changes in model parameters due to the inclusion of specific samples, based on the model's final parameters (Koh & Liang, 2017; Pruthi et al., 2020; Schioppa et al., 2022). However, this approach does not account for the so-called "butterfly effect" in neural networks (Basu et al., 2020; Ferrara, 2024), where omitting a single sample at the beginning of training can alter the gradient trajectory and lead to significant misestimation of a sample's true influence; recent work has shown that influence functions are poor substitutes for true leave-one-out (LOO) retraining (Schioppa et al., 2024; Bae et al., 2022).

Our work's use of the KL-divergence is closely related to the information-theoretic notion of *algorithmic stability*, which investigates the change in a model due to single sample differences in underlying datasets (Bassily et al., 2016; Raginsky et al., 2016; Steinke & Zakynthinou, 2020). For instance, Bassily et al. (2016); Steinke & Zakynthinou (2020); Feldman & Steinke (2018) define *max* KL *stability* and the *average leave one out* KL *stability* as $\sup_{D,D^{-i}} \mathsf{KL}(p_A(\theta \mid D) \parallel p_A(\theta \mid D^{-i}))$ and $\sup_D \frac{1}{n} \sum_{i=1}^n \mathsf{KL}(p_A(\theta \mid D) \parallel p_A(\theta \mid D^{-i}))$ respectively, with $p_A(\theta \mid D)$ being the output distribution over the parameters $\theta$ of an algorithm $A$ trained on the dataset $D = \{z_1, ..., z_n\}$ and $D^{-i} = D \setminus \{z_i\}$. While max KL stability is defined across all possible datasets, average KL stability is defined on a *fixed* underlying dataset. LSI is thus closely related to *average per-sample level KL stability*; this definition is also used by Harutyunyan et al. (2021). Moreover, a connection between LSI and Differential Privacy exists, which we investigate in Appendix H.

**Estimating Parameter Distributions** Computing the KL divergence on the constant sets of parameters (i.e. degenerate random variables) outputted by most (non-Bayesian) learning algorithms is impossible. Negrea et al. (2019); Ye et al. (2023) circumvent this issue by leveraging a Bayesian learning setting, which generates a distribution of parameters, while Rammal et al. (2022) add noise to the final parameters (smoothing) to generate a distribution. Another line of work aims to approximate the training of a neural network by employing continuous stochastic differential equations (Li et al., 2017) or through neural tangent kernels using a linear approximation of a KL divergence-based informativeness notation (Harutyunyan et al., 2021). Our approach mirrors the Bayesian strategy, but we establish a distribution on the network parameters post-hoc by employing a *Laplace approximation* to the posterior parameter distribution (Daxberger et al., 2021; MacKay, 1992).

**Sample Difficulty** As demonstrated by Ethayarajh et al. (2022), sample difficulty is related to sample informativeness, as it (informally) measures the probability that a neural network can learn a sample. Previous studies approximate sample difficulty by examining the impact of model compression on individual samples (Hooker et al., 2020b;a) or by assessing the effect of individual samples on the loss of a held-out dataset (Mindermann et al., 2022; Wu et al., 2020). Our work is related to methods that infer sample difficulty from loss landscape geometry (Zielinski et al., 2020; Chatterjee, 2020; Agarwal et al., 2022; Katharopoulos & Fleuret, 2018), as the Laplace approximation leverages curvature information. However, we note that –while sample difficulty can be a proxy for informativeness– the two concepts are distinct, and, while the aforementioned methods use sample difficulty to infer informativeness (Agarwal et al., 2022), we directly measure sample informativeness, which allows us to draw conclusions about sample difficulty.

## 3 LAPLACE SAMPLE INFORMATION

Next, we formally introduce LSI and its theoretical underpinnings.

**Sample Information** As discussed above, LSI is a per-sample informativeness notion based on the KL divergence, the arguably most established information theoretic divergence. Consider a (probabilistic) training algorithm $A$, a training dataset $D = \{z_1, ..., z_n\}$, an input-label pair of interest $z_i = (x_i, y_i)$, and the training dataset with $z_i$ removed $D^{-i} = D \setminus \{z_i\}$. The distribution of parameters induced by $A$ is denoted as $p_A(\theta \mid D)$, where $\theta \in \mathbb{R}^K$ is a specific realization of the random variable of the parameters $\Theta$. To approximate the (per-sample) information flow of the sample of interest $z_i$ to the neural network's parameters, we compute the KL divergence of the distribution of parameters of a model trained on $D$ to the distribution of parameters of a model trained on $D^{-i}$. This information measure is called the *Sample Information* (SI) of $z_i$:

**Definition 1** (Sample Information). The *Sample Information* of $z_i$ is defined as:

$$\mathsf{SI}(A, D, z_i) = \mathsf{KL}\left(p_A\left(\theta \mid D\right) \parallel \left(p_A\left(\theta \mid D^{-i}\right)\right)\right).$$

SI is justified information-theoretically, as it represents an upper bound on the point-wise conditional mutual information between the parameters of the trained neural network and the datapoint $z_i$. For this reason, SI has already been used in previous literature (Harutyunyan et al., 2021).

Alternatively, SI can be understood through the lens of information-theoretic hypothesis testing: In information theory, $\mathsf{KL}(p_A(\theta \mid D) \parallel p_A(\theta \mid D^{-i}))$ can be interpreted as the *expected discrimination information* or the *expected weight of evidence* of $D$ vs. $D^{-i}$ through observing $\theta$. In other words, SI can be interpreted as the weight of the discriminatory evidence induced by $z_i$. Intuitively, samples that contribute a lot of information to $\theta$ allow for improved insight about the underlying dataset compared to samples that contribute little information, which the KL divergence can measure.

**Laplace Approximation** The definition of SI above implies that the training algorithm is probabilistic; however, most neural networks are trained with Maximum Likelihood Estimation (MLE). This does not result in a distribution over trained parameters which is required to estimate the KL divergence. To remedy this, previous works have leveraged probabilistic/Bayesian neural network training (Negrea et al., 2019; Ye et al., 2023) which can be computationally expensive, or added noise directly to the final parameters (Rammal et al., 2022), which reduces model utility. In contrast, we propose to construct a *quasi-Bayesian* learner from the trained model parameters by fitting a posterior probability distribution to the parameters using a Laplace approximation, a classical technique in Bayesian inference, whose application in neural networks has recently witnessed increased interest.

We use the "classical" Laplace approximation (Daxberger et al., 2021) throughout and discuss the recently proposed *Riemann* Laplace approximation (Bergamin et al., 2024) in Appendix I. The Laplace approximation leverages a second-order Taylor expansion of the loss landscape around the final parameters $\theta$ of the model $\mathcal{A}_\theta$. It thus assumes that the loss function is convex around the local minima and can be approximated by a quadratic function, which is a common assumption made for the analysis of the training behavior of neural networks (Wen et al., 2020; Zhu et al., 2018; He et al., 2019). The Laplace approximation models the parameters of the neural network as multivariate Gaussian distributions whose covariance is based on the local curvature of the loss landscape. This normality assumption is made by invoking the central limit theorem on the distribution of the parameters after many training steps. Despite these assumptions, the aforementioned works show the Laplace approximation to be efficient and competitive with fully Bayesian training in deep networks, which motivates its use in our work.

Formally, suppose that a neural network is trained on a dataset $D$ by minimizing the loss function $\mathcal{L}(D, \theta)$ with $l(x_n, y_n; \theta)$ as the empirical loss term and regularizer $r(\theta)$ on the parameters $\theta \in \mathbb{R}^K$ resulting in:

$$\hat{\theta} = \arg \min_{\theta \in \mathbb{R}^K} \mathcal{L}(D, \theta) = \arg \min_{\theta \in \mathbb{R}^K} \left( r(\theta) + \sum_{n=1}^N l(x_n, y_n; \theta) \right). \tag{1}$$

Considering the empirical loss term as an i.i.d. log-likelihood $l(x_n, y_n; \theta) = -\log p(y_n \mid \mathcal{A}_\theta(x_n))$ and the regularizer as a log-prior $r(\theta) = -\log p(\theta)$ establishes $\hat{\theta}$ as a *maximum a-posteriori* estimate (MAP) of the parameters. In other words, the MLE of the regularized learner is interpreted as a MAP solution whose prior is defined by the regularizer. The Laplace approximation then uses a second-order Taylor expansion of $\mathcal{L}$ around $\hat{\theta}$ to construct a Gaussian approximation of $p_A(\theta \mid D)$:

$$\mathcal{L}(\mathcal{D}, \theta) \approx \mathcal{L}(\mathcal{D}, \hat{\theta}) + (\nabla_\theta \mathcal{L}(D, \hat{\theta}))(\theta - \hat{\theta}) + \frac{1}{2}(\theta - \hat{\theta})^T (\nabla_\theta^2 \mathcal{L}(D, \hat{\theta}))(\theta - \hat{\theta}). \tag{2}$$

Considering a converged model, the first order term vanishes as $\nabla_\theta \mathcal{L}(D, \theta)|_{\hat{\theta}} \approx 0$ yielding the Laplace approximation of the posterior distribution as:

$$p(\theta|D) \approx \mathcal{N}(\hat{\theta}, \Sigma) \text{ with } \Sigma = (\nabla_\theta^2 \mathcal{L}(D, \hat{\theta}))^{-1}. \tag{3}$$

Remark that while the derivation of the Laplace approximation requires the model to be converged, we show that the sample order established by LSI remains largely consistent across training. Thus, according to our empirical findings in shown Appendix G, employing LSI to gain insights does not necessitate the convergence of the model.

**Laplace Sample Information (LSI)** After introducing the key theoretical insights, we are now ready to define LSI. Concretely, we combine the Laplace approximation of the posterior parameter distribution (Equation (3)) and the definition of sample informativeness (Definition 1), which yields our definition of *Laplace Sample Information*:

---

**Definition 2** (Laplace Sample Information). Let $A$ be a (non-Bayesian) learning algorithm with parameters $\theta$, a loss function $\mathcal{L}$, a dataset $D$ and $D^{-i} = D \setminus \{z_i\}$; moreover, let $\theta_{\mathrm{MLE}}$ be the *maximum likelihood estimate* of $\theta$. Then, the LSI of the investigated sample $z_i$ with respect to $A$ and $D^{-i}$ is defined as:

$$\mathsf{LSI}(z_i, A, D^{-i}) \triangleq \mathsf{KL}(\mathcal{N}(\hat{\theta}, \Sigma) || \mathcal{N}(\hat{\theta}^{-i}, \Sigma^{-i})). \tag{4}$$

Above, $\hat{\theta} \triangleq \theta_{\mathrm{MLE}}(A(D))$ and $\hat{\theta}^{-i} \triangleq \theta_{\mathrm{MLE}}(A(D^{-i}))$.

Moreover, $\Sigma \triangleq (\nabla_\theta^2 \mathcal{L}(D, \hat{\theta}))^{-1}$ and $\Sigma^{-i} \triangleq (\nabla_\theta^2 \mathcal{L}(D^{-i}, \hat{\theta}^{-i}))^{-1}$.

---

The Laplace approximation results in a Gaussian distribution over $K$ neural network parameters, enabling the computation of the KL divergence and, consequently, the LSI as follows:

$$\mathsf{LSI}(z_i, A, D^{-i}) = \frac{1}{2} \left[ \mathrm{tr}((\Sigma^{-i})^{-1} \Sigma) - K + (\hat{\theta}^{-i} - \hat{\theta})^T (\Sigma^{-i})^{-1} (\hat{\theta}^{-i} - \hat{\theta}) + \ln \left( \frac{\det(\Sigma^{-i})}{\Sigma} \right) \right].$$

LSI has a notable benefit: since the Laplace approximation is computed *after* model training, the computation of LSI is agnostic to model architecture, the dataset, the loss function, and the learning

setting. In Appendix A, we show the benefits of LSI compared to two other information measures from recent literature (Harutyunyan et al., 2021; Wongso et al., 2023).

**Efficient Computation of LSI**   In terms of computational considerations, the Laplace approximation of a neural network requires the computation of the Hessian of the loss function with respect to the parameters. Unfortunately, for $K$ total model parameters, the Hessian requires $\mathcal{O}(K^2)$ memory, which is intractable even for moderate networks. Additionally, the inversion of the Hessian for computing the covariance matrix has a time complexity of $\mathcal{O}(K^3)$. To address this issue in practice, we leverage the established technique of using a Hessian approximation. We note that LSI is agnostic to the exact approximation. For example, block-diagonal approximations such as K-FAC or their eigenvalue-based counterparts (Grosse & Martens, 2016; Bae et al., 2018) can be used. However, in our experiments, we find that even a diagonal approximation works very well, i.e. discarding all but the diagonal Hessian entries (Farquhar et al., 2020; Kirkpatrick et al., 2017), resulting in $\mathcal{O}(K)$ in memory and a time complexity of only $\mathcal{O}(K)$ for the inversion. We show that the LSI of this diagonal approximation (as well as the KFAC approximation) is well-correlated with the LSI using the full Hessian in Appendix C, indicating the applicability of the diagonal approximation.

## 4  EXPERIMENTS

### 4.1  EXPERIMENTAL SETUP

**Model and Datasets**   We demonstrate the utility of LSI with experiments on supervised image tasks using CIFAR-10, CIFAR-100 (Krizhevsky & Hinton, 2009), a medical imaging dataset (pediatric pneumonia, i.e. lung infection in children), (Kermany et al., 2018) and two ten-class subsets of the ImageNet dataset (Deng et al., 2009), *Imagewoof* and *Imagenette* (Howard, 2019). We select CIFAR-10 and CIFAR-100 as easy and challenging benchmarks, and the two ImageNet subsets as easy and challenging representatives of large image datasets. The pneumonia dataset represents a challenging real-world use case, as two of the classes (bacterial and viral pneumonia) are hard to distinguish, even for human experts.

Recall that LOO-estimation of LSI for any given sample necessitates retraining a model. To circumvent this requirement and lessen the computational burden, we employ a *probe*, whereby we estimate LSI on a small model trained on image features computed by a frozen feature extractor. For the image tasks, we employ a ResNet-18 (He et al., 2016) up to the fully connected layer pre-trained on ImageNet as the frozen *feature extractor* and append a single hidden layer with a ReLu activation function followed by a classification layer as our task-specific classification head/ probe model. The fact that the sample informativeness order captured by LSI on the probe transfers excellently to larger models, which is discussed in detail in Section 4.4 and Appendix D, is one of LSI's key benefits. We find that employing a probe allows the computation of the sample information with a speedup of at least three orders of magnitude. Interestingly, despite the fact that the Laplace approximation requires a converged model, empirically LSI is applicable even far prior to model convergence (after a brief warm-up period), as shown in Appendix G.

Beyond image classification, we show the applicability of LSI in a text sentiment analysis task on the IMDb dataset (Maas et al., 2011). As a model, we employ a pre-trained BERT (Devlin et al., 2019) as a feature extractor and the probe model described above.

Moreover, we compute LSI in unsupervised contrastive learning using CLIP (Radford et al., 2021) between image and caption pairs of the COCO dataset (Lin et al., 2015). To decrease the computational complexity, we reduce the dataset to images carrying segmentation masks of bananas or elephants. We use frozen vision and language transformers (BERT (Devlin et al., 2019) and ViT-B/16 (Dosovitskiy et al., 2020)) as a backbone and compute LSI with respect to the embedding layer (acting as the probe).

To evaluate LSI independently of the sample order, we used full-batch gradient descent for model training. Note that LSI makes no assumptions about the training process and remains applicable in a batched training setting, either combined with multiple re-training to average out the effect of batch sample ordering or sampling from the entire dataset, e.g. using Poisson sampling (Abadi et al., 2016).

**Training Parameters and Hardware**   All training (hyper-)parameters are provided in Appendix N and a description of the hardware used and resources required is provided in Appendix O.

## 4.2 ESTIMATING DATA TYPICALITY, INFORMATIVENESS AND DIFFICULTY USING **LSI**

**LSI Orders Samples According to Typicality** We begin by showing that LSI provides an interpretable notion of individual sample informativeness. Recall that, as shown by Feldman (2020), data-generating distributions tend to exhibit a *long tail* concerning sample typicality (i.e. view angle, object size, etc.). LSI unveils this long-tailed distribution, as the LSI distribution on the examined datasets also exhibits a long tail (Figure 1). As most samples have low LSI and only a few samples have high LSI, only a small subset of the data is highly informative, while most samples contribute little information to the parameters. Note that, as the model has no prior "knowledge" about typicality or about the data distribution, this notion of sample typicality arises during training and is encoded through the amount of information individual samples contribute to the model. Thus, this notion of typicality does not necessarily coincide with its human interpretation.

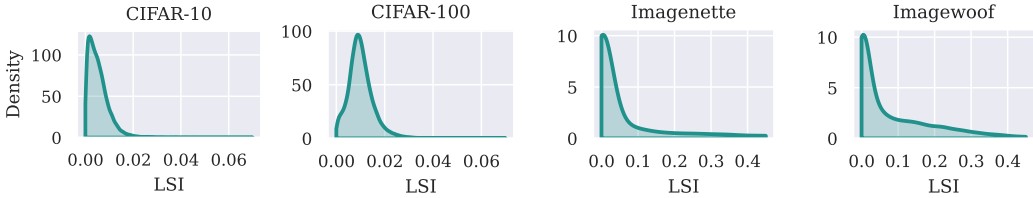

Figure 1: LSI distribution across individual samples of the investigated datasets.

Investigating samples with low and high LSI, we find that samples that are strongly representative of their class provide little unique information to the model parameters. Consequently, representative samples have low LSI, while samples with high LSI often are cropped, mislabeled, or show the object from an uncommon perspective. This indicates that the model "encodes" the long-tailed sub-population distribution described by Feldman (2020).

For example, while the least informative samples in the *Australian terrier* class in ImageNet are well-framed and highly typical representatives of the breed, high LSI samples also contain a variety of different dog breeds, i.e. mislabeled samples. Similarly, high LSI samples of the *radio* class contain an out-of-distribution image of a car. Low LSI samples in the *pneumonia* dataset contain chest x-rays that have similar exposure and framing, while high LSI samples contain cropped, over/ underexposed images as well a chest-x-ray of an adult woman, which is clearly wrongly included in a pediatric (children's) dataset (Figure 2).

A larger selection of images beyond the aforementioned representative findings can be found in Appendix K (ImageNet), Appendix L (CIFAR-10), and Appendix M (pneumonia dataset). Exemplarily, in CIFAR-10, the lowest LSI within the *bird* and *airplane* categories feature a blue sky background, while the samples carrying the highest LSI have the sun as the background for *airplanes* or are frontal full-frame views of ostriches, a rather unrepresentative member of the *bird* class.

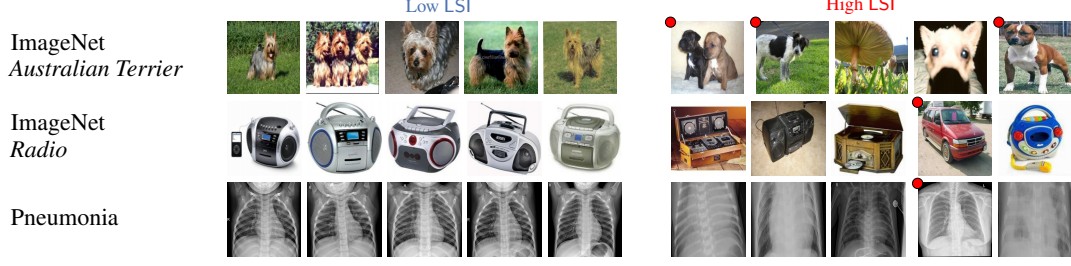

Figure 2: Selected images with low/high LSI in ImageNet and the medical dataset. Samples with low LSI are representative of their underlying class, whereas high LSI samples are often mislabeled/ out-of-distribution (red dots) or atypical with respect to exposure, viewing angle, etc.

In the text domain, unambiguous texts aligning with their label carry low sample informativeness and thus are assigned low LSI. Conversely, high LSI indicates mislabeled, ambiguous, or label-contradicting texts, as shown exemplarily in Figure 3.

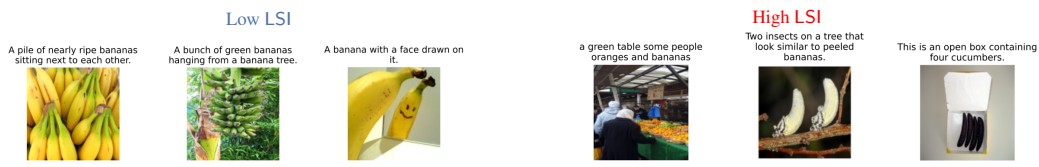

**Low LSI sample with positive label:**
This film **is brilliant**! It **touches everyone who sees it in an extraordinary way**. It really takes you back to your youth and puts a new perspective on how you view your childhood memories. There are **so many layers to this film**. It is **innovative and absolutely fabulous!**

**Legend:** *Ambiguity* – Label contradicting – **Label confirming**

**High LSI sample with negative label:**
*Early, heavy,* **war-time propaganda** *short urging people to be careful with their spending practices [...] Using* **scare**, **guilt** *and [...]* **war-time** *production and therefore wage increase and subsequent spending practices [...]* **serious problems** *during and after the war.* It truly is a window into the past, historically and culturally.

Figure 3: LSI in supervised text classification on the IMDb dataset using BERT

Even in multi-modal tasks, we find that typical samples with a prominent portrayal of class representatives (e.g. *bunches of bananas*) have low LSI, as they carry little unique information. Images in which the subject is not prominently shown, not corresponding to the *banana* label, or do not contain any bananas (e.g. *box of cucumbers*) have high LSI (shown in Figure 4).

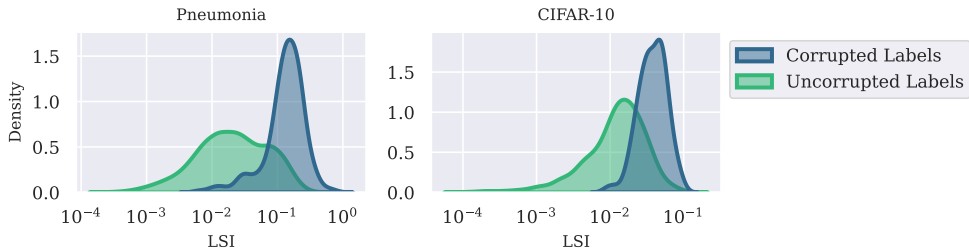

Low LSI — High LSI

A pile of nearly ripe bananas sitting next to each other. — A bunch of green bananas hanging from a banana tree. — A banana with a face drawn on it. — a green table some people oranges and bananas — Two insects on a tree that look similar to peeled bananas. — This is an open box containing four cucumbers.

Figure 4: Selected images with low/high LSI of contrastive learning on COCO

**Focusing on Mislabeled Samples**  To further show the effect of mislabeled data on sample informativeness and demonstrate the capabilities of LSI in detecting mislabeled samples, we apply deliberate label noise (random label flips to any other class) to 10% of CIFAR-10 and pneumonia dataset samples and recompute LSI for all samples. Figure 5 shows that mislabeled samples, on average, have substantially higher LSI and thus provide more unique sample information. Notably, a portion of mislabeled samples has higher LSI than even the most informative correctly labeled samples, indicating the disproportionate detrimental effect of poor quality data and highlighting the importance of meticulous dataset curation (Zha et al., 2023). Further experiments on human mislabeled data are in Appendix E.

Figure 5: LSI distribution on data with corrupted labels (mislabeled) vs. uncorrupted labels

LSI can furthermore effectively distinguish label corruption (mislabeling, Figure 6 left) from data corruption (by including non-informative *lorem ipsum* text with random labels, Figure 6 right). Since *lorem ipsum* is self-similar and provides little information (neither label confirming nor contradicting), the LSI is concentrated around an intermediate value, whereas the mislabeled text has higher LSI, as it is more informative to the model (strong contradictory information).

**Sample Informativeness Increases with Smaller Datasets**  Interestingly, the LSI for the ImageNet subsets is about one order of magnitude larger than for CIFAR-10 when using the same experimental setup (same training parameters and model), as seen in Figure 1. Due to the smaller dataset size (9 469 compared to 50 000), individual samples of the ImageNet subsets provide more information to the neural network than individual samples of CIFAR-10. This result is corroborated by picking a subset of the *same* dataset (see Appendix F). Therefore, with smaller datasets, the presence or absence of individual samples becomes more influential on the model parameters.

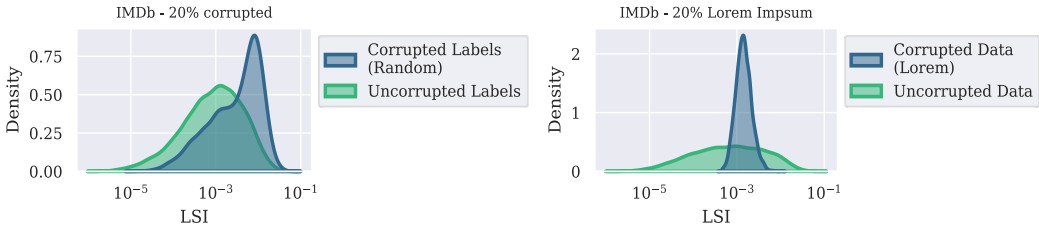

Figure 6: LSI in supervised text classification on the IMDb dataset using BERT

**Establishing a Class-wise Informativeness Ordering using LSI**  We next investigate the *class-wise* distribution of LSI, which allows for reasoning about which class imparts the most information to the model parameters during model training. Figure 7 (left) shows the class-specific LSI for the three different classes of the pediatric pneumonia dataset. We consulted a diagnostic radiologist who confirmed that this class ranking corresponds to the difficulty humans have in classifying these pathologies in children. In particular, while classifying a radiograph as *abnormal* is easy, distinguishing between *bacterial* and *viral* pneumonia is difficult, which mirrors the relationship between the distributions in the figure. To further emphasize the capabilities of LSI to assess class informativeness, we investigate the *change* in class-level distributions of LSI on the pneumonia dataset when the feature embeddings of one class are corrupted with additive Gaussian noise (variance as *data noise* in Figure 7). Note that adding noise to the embeddings increases variance and makes the samples seem "less similar". This is different from Figure 6 right, where the corrupted data is highly self-similar and has little effect on LSI. Here, since the model is required to exploit more of the information contained in every sample, the individual samples become more informative; correspondingly, with higher added noise, the LSI distribution of the corrupted class shifts towards higher values.

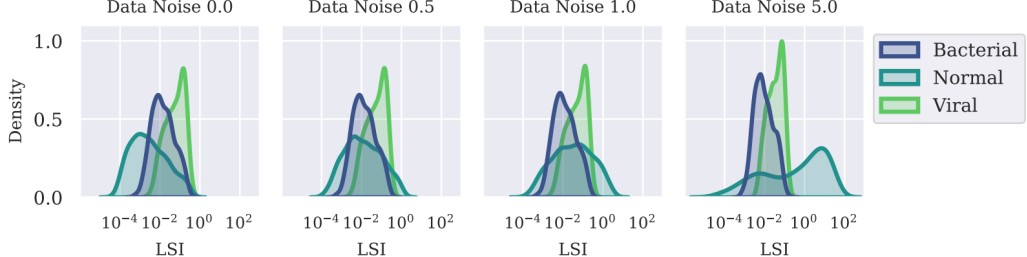

Figure 7: LSI distributions across three classes of the pneumonia dataset. By increasing the additive Gaussian noise standard deviation (scaled to the initial standard deviation of the embedding) on the *normal* class, their dissimilarity increases, requiring the model to extract more of their unique information in its parameters. This increase in informativeness leads to higher class-wise LSI.

**Measuring Dataset Difficulty using LSI**  Comparing the LSI distribution between datasets allows us to reason about the overall dataset's difficulty. For example, samples of CIFAR-100 have, on average, higher LSI than the samples of CIFAR-10 when using the same experimental setup (see Figure 1). This indicates that CIFAR-100 contains more samples that carry higher unique information than CIFAR-10 which aligns well with CIFAR-100 having more classes and thus fewer samples per class than CIFAR-10 and being generally considered more difficult. The same phenomenon arises when comparing the distributions of the ImageNet subsets with the easier Imagenette having a lower average LSI and a slimmer tail than the more difficult Imagewoof.

### 4.3 LSI AS A PROXY FOR SAMPLE LEARNABILITY AND OUT-OF-SAMPLE ACCURACY

**Sample Learnability**  As discussed above, sample *difficulty* has been previously used as a proxy for sample informativeness (Agarwal et al., 2022). We next show that this relation is indeed justified by

demonstrating that the "reverse direction" also holds, i.e. that sample informativeness also predicts sample difficulty, which we define as the sample-wise training accuracy. To this end, we partition each dataset into label-stratified subsets of $1/3$ of the full dataset size. Note that, since choosing the subsets based on LSI alone may introduce class imbalances, as classes are not uniformly distributed with respect to their LSI, we construct *stratified* subsets by combining data with respect to their *class-specific* LSI orderings. For instance, the subset containing samples with the highest LSI contains the $1/3$ of samples of each class carrying the highest LSI. While we show the results on CIFAR-10 here, these results generalize to all other investigated datasets, which are shown in Appendix J.

Following the hypothesis that more informative samples are harder to learn (i.e., classify accurately), we define sample *learnability* as the probability that a model correctly assigns the label during training (equivalently to top-1-accuracy). We observe that the learnability follows the ordering of the subsets with respect to their LSI (Figure 8 left): models trained on a subset of low LSI samples achieve substantially higher training accuracy than models trained on subsets with medium or high LSI. Thus, LSI effectively serves as a means of ordering samples with respect to their learnability for the model.

**Out-of-Sample Accuracy** While learnability assesses *training* accuracy, that is, the ability of the model to *fit*, it is also (or perhaps more) important to assess how valuable the information gleaned from training samples is for predicting unseen samples, i.e. out-of-sample (*test*) accuracy.

Regarding test accuracy (Figure 8 right), the models trained on intermediate LSI data exhibit the best test accuracy and smallest train/test gap. Samples with low LSI have (near-)perfect train accuracy but reduced test accuracy, while high LSI samples are not fitted well (low train accuracy) but tend to generalize better than their training performance. This indicates that intermediate and high LSI samples form the dataset partitions generally associated with learning generalizable representations and benign memorization (as defined by Feldman (2020)). In contrast, low LSI samples seem to be disposable or even harmful to generalization (i.e., the model overfits on sets of low LSI samples).

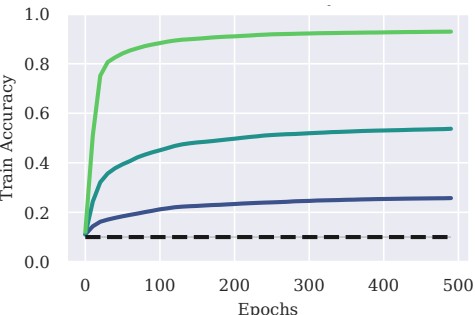 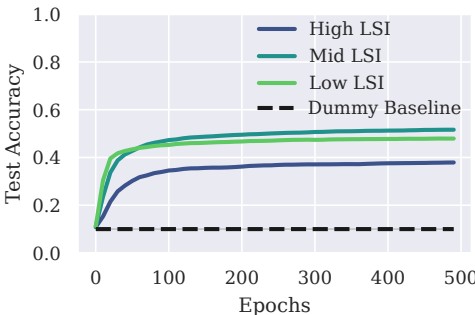

Figure 8: Training accuracy (left) and test accuracy (right) of models trained on subsets of CIFAR-10 containing $1/3$ of the complete dataset with the highest, intermediate, and lowest LSI values compared to the dummy baseline of a model predicting the majority class.

## 4.4 TRANSLATING LSI BETWEEN PROBE AND FULL MODEL

So far, we considered the LSI computed on a probe consisting of a fully connected layer acting on the features computed offline by a frozen ResNet-18. This allows for the computation of the LSI for each sample in the dataset in a short amount of time while keeping the respective Hessian tractable. LSI allows the establishment of an ordering of samples with respect to the information they impart in the model parameters. As shown previously, this orders the data from the most typical samples of a respective class (on which the probe model overfits) to the most atypical samples (which the probe is incapable of fitting well). As LSI is a sample-specific property, we expect an (approximately) consistent ordering of samples when training a model directly on the underlying data (rather than on the feature embeddings). To investigate this hypothesis, we repeat the experiment described in Section 4.3 on CIFAR-10 by training a ResNet-9 on the dataset partitions established by the probe. As shown in Figure 9, the same pattern emerges as when the probe is used concerning the sample learnability and generalization capabilities: the low LSI subset is easy to fit but generalizes just slightly worse than the mid LSI subset, and the high LSI subset is hard to fit and the worst in terms of test accuracy. However, as the ResNet-9 model has more learnable parameters compared to the probe

model, it is capable of overfitting on the high LSI data. Interestingly, overfitting on high LSI samples occurs later in training than overfitting on low/intermediate LSI samples. The sample informativeness ordering as derived by the probe generalizes to other datasets and other model architectures and is consistent. Further results are shown in Appendix J. Additionally, this ordering emerges very early during training and does, in fact, not require the model to be fully converged (Appendix G).

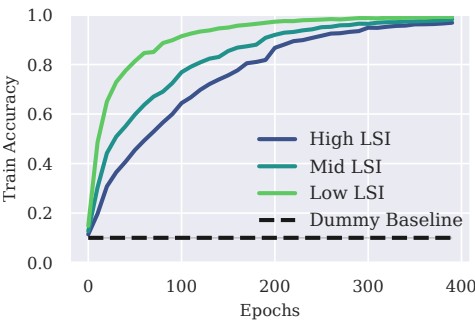 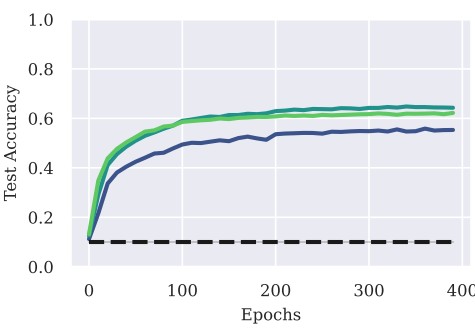

Figure 9: Training accuracy (left) and test accuracy (right) of a ResNet-9 trained (from scratch) on subsets of CIFAR-10 containing $1/3$ of the complete dataset with the highest, intermediate and lowest LSI values and compared to the dummy baseline of a model always predicting the majority class.

## 4.5 ADDITIONAL FINDINGS AND RESULTS

We refer to the Appendix for a comparison of LSI with other methods (Appendix A, Appendix B) and additional experiments investigating Hessian approximations (Appendix C); the correlation between LSI from probing vs. full models (Appendix D); effect of humanly mislabeled data on LSI (Appendix E); effects of dataset size on LSI (Appendix F); the robustness of LSI throughout training (Appendix G); its relationship with Differential Privacy (DP) (Appendix H); its computation via the Riemann Laplace approximation (Bergamin et al. (2024), Appendix I), transferring LSI from the probe to other architectures (Appendix J), as well as for a collection of samples with low and high LSI for selected datasets (Appendices K to M).

## 5 DISCUSSION AND CONCLUSION

We introduced *Laplace Sample Information*, a measure of information flow from individual samples to the parameters of a neural network, i.e. of the information a training sample contributes to the final model. We demonstrated the capabilities of LSI as a tool to predict typical/atypical data, with atypical samples carrying more unique information, and for detecting out-of-distribution and mislabeled samples. Furthermore, we established LSI as a class-wise informativeness measure, which correlates with human perception and underscores the utility of LSI for reasoning about dataset difficulty. Therefore, we expect LSI to be a useful measure of sample-wise informativeness in various areas of ML research. Compared to recently proposed informativeness measures, LSI does not require linearization and is much more scalable than Harutyunyan et al. (2021). Moreover, it applies to arbitrary architectures and task-agnostic (e.g. to self-supervised learning with CLIP) contrary to Wongso et al. (2023). For a detailed comparison to these techniques, see Appendix A.

Due to the high computational cost of LOO retraining, we employ a probe to compute LSI. Nonetheless, the ordering properties defined by LSI on the probe translate effectively to larger models. We note that using a probe as a proxy remains a design choice favouring efficiency; the LSI formalism is model-agnostic and can be used with any Hessian approximation. As shown in Appendix C, LSI can be used with memory-efficient (K-FAC); we thus anticipate that it can transfer to LLM-scale workflows (Grosse et al., 2023) which we intend to investigate in the future.

In conclusion, LSI combines model- and data-centric approaches to estimate sample informativeness. We foresee an important role in combining metrics like LSI with characterizations determined through human reasoning to facilitate model and dataset introspection, for example, in topics like algorithmic fairness or AI safety.

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

## A  COMPARISON TO OTHER INFORMATIVENESS METRICS

As our work is an approximation of point-wise mutual information, we compare LSI with point-wise mutual information measures and bounds thereof. Since mutual information is hard to compute in high-dimensional settings such as deep learning, "slicing" has gained traction as a scalable and efficient method for computing it, which retains some of the properties of the original MI (Goldfeld & Greenewald, 2021). We thus compare LSI to point-wise sliced mutual information (PSMI) (Wongso et al., 2023) and also to smooth unique information (SUI) (Harutyunyan et al., 2021) Figure 10. To our knowledge, these are the only methods performing point-wise informativeness estimation based on mutual information, that claim to scale beyond toy datasets. We thus believe that these are the only salient comparisons to other techniques from current literature.

Our comparison shows that, while SUI and PSMI are nearly uncorrelated, likely due to compounding approximation error, LSI is strongly correlated with PSMI and weakly with SUI, indicating that – while both SUI and PSMI measure some aspect of sample informativeness – neither of them seems to capture the full picture. LSI is positively correlated with PSMI, which is a result of the relationship between the KL divergence and point-wise mutual information mentioned in Section 3. Interestingly, the fact that sliced MI is a computational approximation of mutual information and lacks some of the guarantees inherent to the "true" MI (notably, the absence of the information-processing inequality) seems to somewhat diminish PSMI's expressiveness. LSI, on the other hand, seems to suffer less from the approximation of the Hessian (see Appendix C). Thus, LSI and PSMI measure distinct but related quantities, and both detect the same underlying structure in the data. In terms of SUI, the metric relies on a global linearization of the entire network, a drastic intervention that can make the model fragile (Ortiz-Jimenez et al., 2021). Moreover, SUI is extremely memory-inefficient and does not scale to larger datasets (the SUI paper limited its evaluation to binary classification datasets with ca. 1000 samples). LSI is much more flexible regarding the choice of Hessian approximation and Laplace approximation, does not intervene on the model weights, and does not have the same memory inefficiency problem. Our comparison is thus based on a 1000 sample three class subset classification task of CIFAR-10.

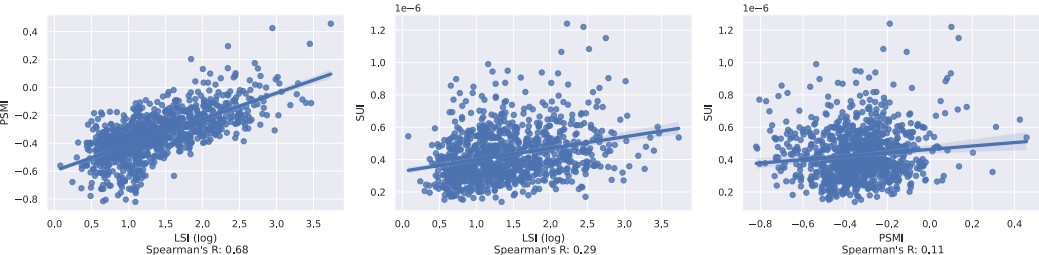

Figure 10: Correlation plots between LSI, Smooth Unique Information (SUI) and Point-wise Sliced Mutual Information (PSMI) on a CIFAR-10 subset. While LSI and PSMI are strongly correlated, SUI and LSI exhibit a weaker correlation, likely due to the limitations of SUI. SUI and PSMI are nearly uncorrelated, likely due to compounding approximation errors.

## B  COMPARISON OF LSI WITH TRAK

To show the differences between leave-one-out retraining-based data valuation metrics to influence estimation, we compare LSI with TRAK (Park et al., 2023) and discuss the results. Note that while both methods perform data attribution, they differ in their fundamentals. LSI measures the informativeness of individual samples to the weights of the trained neural network. At the same time, TRAK aims to measure the influence of individual samples on correctly predicting a sample of interest. Therefore, LSI provides a singular value for each sample in the train set, while TRAK provides a $N \times M$ tensor of values with $N$ and $M$ being the sample count of the train and test set, respectively. To compare these two measures we assign the datapoints in the train set the average of their TRAK values across all test datapoints.

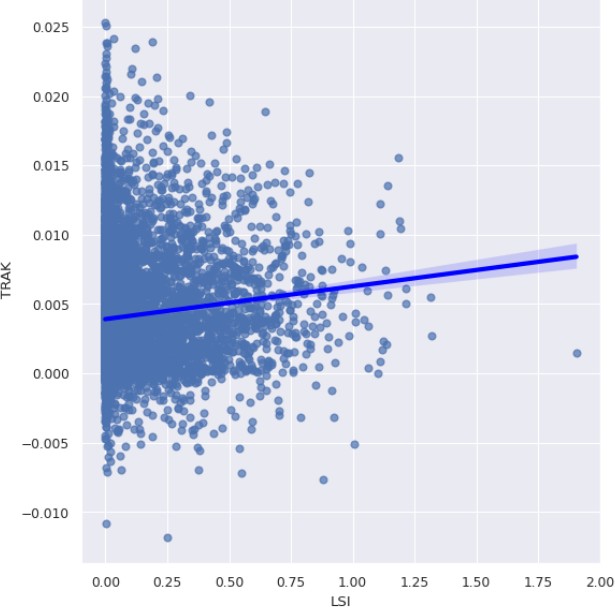

Figure 11: Correlation with confidence interval of 0.95 of LSI with TRAK

Figure 11 shows a weak correlation (Spearman's Rank correlation of 0.15) between LSI and TRAK scores assigned to each datapoint in the Imagenette dataset.

To further compare LSI and TRAK, we perform a qualitative analysis of samples assigned the highest and lowest values of these measures. Figure 12 shows that LSI separates the dataset into samples of little unique information (less typical) carrying low LSI and samples of large unique information (visually dominated by people, mislabeled or out of distribution) of high LSI. Opposingly, TRAK fails at this, with out of distribution images like the car carrying average TRAK scores. Moreover, the images of low and high TRAK scores do not substantially differ, as both show clear images of radios, however with white background for high TRAK scores. Therefore, we conclude that while TRAK may be meaningful in assessing sample-wise cross-influence, it fails as a data attribution measure specific to a sample.

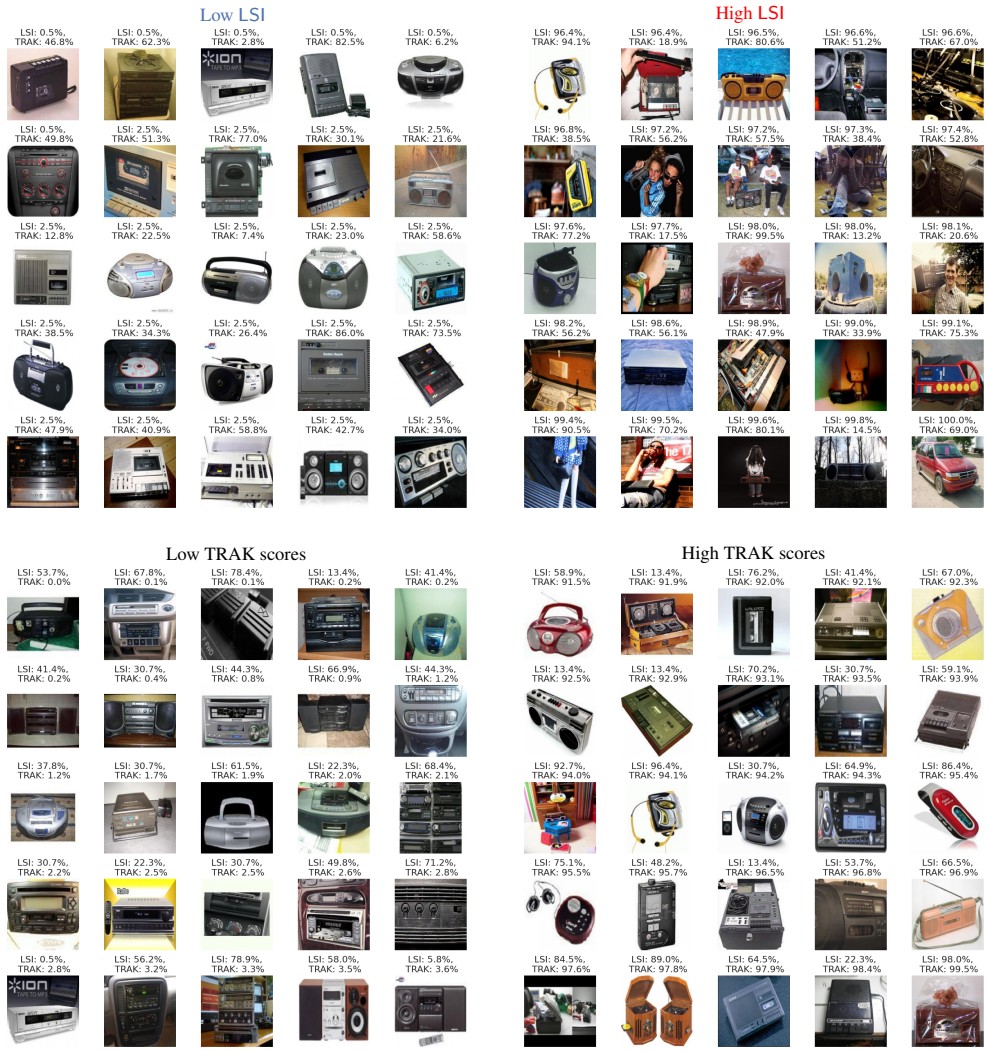

Figure 12: Exemplary images of the highest and lowest LSI and TRAK for the *Radio* class in Imagenette. The images are labeled with corresponding percentiles of them in the measure-ordered dataset.

# C APPROXIMATION OF THE HESSIAN

Due to computational constraints, we refrain from using the full Hessian of the parameters of the neural network to compute the LSI and use a diagonal approximation. Figure 13 shows that LSI computed on the diagonal approximation of the Hessian, LSI computed on the KFAC approximation of the Hessian and LSI computed on the full Hessian correlate strongly with a Spearman's $R > 0.9$.

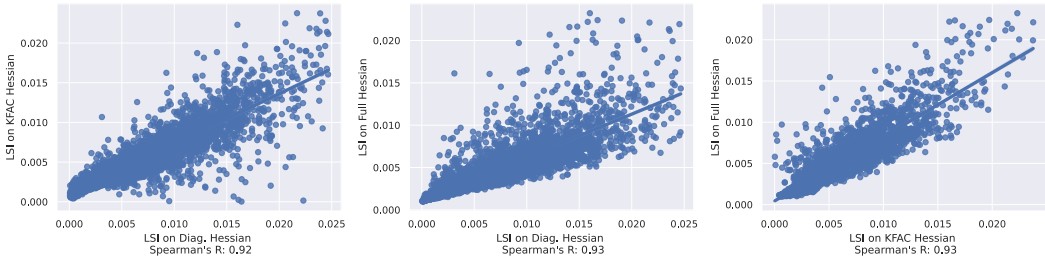

Figure 13: Correlation between LSI on the full Hessian, on the diagonal approximation and the KFAC approximation of the Hessian. All approximations correlate strongly with LSI computed on the full Hessian. LSI is computed using the last-layer probe on CIFAR-10

# D PROBE MODEL APPROXIMATION OF **LSI**

While LSI does not put any constraints on the model it is computed on, training large models until convergence in a leave-on-out setting is computationally very expensive. Thus, we employ a probe model, that acts upon the features probed from a pre-trained feature extractor. To indicate the applicability of this probing approach, we show the correlation of scores computed on 400 samples of CIFAR-10 for LSI computed on all parameters of a CNN (3 convolutional layers with one fully connected layer as head), last-layer parameters of the CNN and the probe model. Note, all except the last require the retraining of the whole model for each data point. Computing LSI on the probe solely requires the retraining of the probe, which can be done in a small fraction of the time. Table 1 shows that LSI computed using the probe model is an excellent approximation to LSI computed on a full CNN with a Spearman's $R > 0.9$.

|  | Probe-Model Parameters | Last-Layer Parameters |
|---|---|---|
| Full CNN Parameters | 0.93 | 0.98 |
| Last-Layer Parameters | 0.90 | - |

Table 1: Spearman's $R$ between LSI computed on all parameters of a CNN, the last layer parameters and a probe model.

## E  HUMANLY MISLABELED EXAMPLES

To extend our experiment on the effect of mislabeled data on LSI, we conduct further experiments on CIFAR-10N with the aggregated labels setting, which assigns the label to a datapoint according to a majority vote across several human labellers (Wei et al., 2021). CIFAR-10N introduces a set of human labels to the original CIFAR-10 data. With this, it also introduces human mislabeling. Other than the previously considered setting of label flipping, these mislabels should be closer to the correct label. Exemplarily, while with label flipping, an image showing a dog on grass and thus initially labeled as a dog is equally likely to be labeled as an airplane or a deer. With CIFAR-10N, the image of the dog will be more likely to be mislabeled as a deer than as an airplane.

Figure 14 (left) shows the distribution of LSI across correctly labeled samples with human labeling error. As with label flipping (Figure 14 (right)), the individual information mislabeled samples contribute to the weights of the model is higher than for correctly labeled samples as measured with LSI.

However, this increase in LSI due to mislabeling is smaller than for label flipping. Consequently, mislabeling due to human misinterpretation results in data-label pairs that provide less unique information than label flipping. This follows our intuition that, exemplarily, a brown figure (a dog) on a green landscape labeled as deer is less unique than if it were labeled as an airplane.

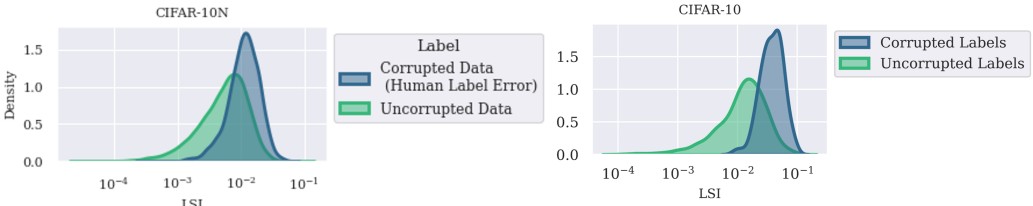

Figure 14: LSI distribution on data with corrupted labels (mislabeled) vs. uncorrupted labels with human mislabeling (left) and label flipping (right)

## F  **LSI** INCREASES WITH SMALLER DATASETS

To substantiate the findings of Section 4.2, we compare the distribution of LSI on CIFAR10 and on a subset of CIFAR10 of $1/5$ the original size (Figure 15). As discussed previously, with a smaller sample count in the individual datasets, the information individual samples contribute to the parameters increases (in this case, by around one order of magnitude).

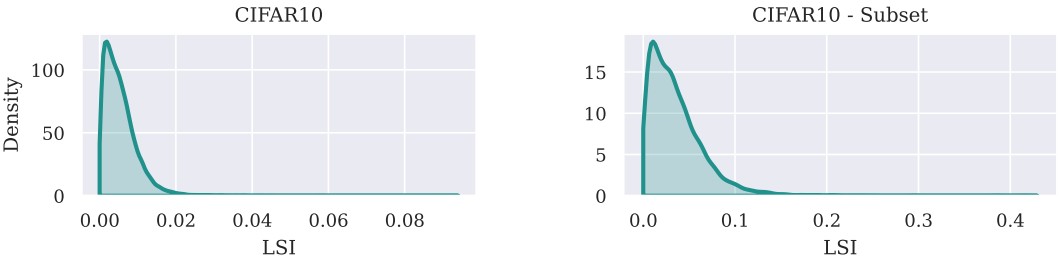

Figure 15: LSI distributions of CIFAR10 and a subset of CIFAR10. The LSI computed on the subset is about one magnitude larger, as with fewer samples, individual samples are required to be more informative to the model.

# G APPLICABILITY OF **LSI** BEFORE MODEL CONVERGENCE

To bound conditional point-wise mutual information, LSI requires the underlying model to be converged. However, in a leave-one-out retraining setting, this can be computationally expensive. We therefore investigate the robustness of sample order as imposed by LSI throughout the training. Figure 16 shows, that after an initial warm-up phase, the sample ordering remains largely consistent. This indicates, that the early stage models approximate the ordering well.

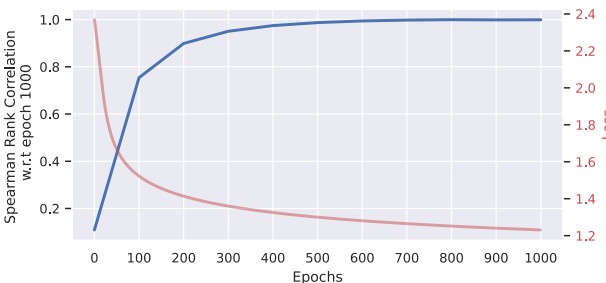

Figure 16: Correlation of the LSI sample ordering between the last epoch (1000) and during training (blue curve). After a warm-up period, the ordering becomes and remains consistent.

# H **LSI** UNDER DIFFERENTIAL PRIVACY

LSI quantifies the information flow from an individual sample to the neural network throughout the training process. DP applied to model training through DP stochastic gradient descent (DP-SGD) (Abadi et al., 2016) offers a means of limiting the influence exerted by any single sample on the parameters of the neural network by clipping and applying noise to the per-sample gradients. LSI lends itself as a natural choice for assessing the impact of individual samples on differentially private training as well as the effect of DP on the contribution of each sample to the neural network's training process, which we investigate here. Note that, as discussed in Section 2, the KL divergence plays an important role in assessing algorithmic stability, and DP implies a strong stability condition, reinforcing this link.

**Differential Privacy Background**  To show the strong connection between LSI and DP, we recall the definition of Rényi differential privacy (RDP) (Mironov, 2017) as:

**Definition 3** (($\alpha, \epsilon$)-RDP). A randomized mechanism $f : \mathcal{D} \to \mathcal{R}$ for $\mathcal{R}$ as the co-domain of $f$ satisfies ($\alpha, \epsilon$)-RDP, if for **all** adjacent $D, D' \in \mathcal{D}$ it holds that

$$\mathsf{D}_\alpha \left( f\left( D \right) \, \| \, f\left( D' \right) \right) \leq \epsilon.$$

Above, the adjacency between $D$ and $D'$ is defined as the two databases differing by a single entry, which can be through adding, removing, or replacing a single sample. RDP is based on the *Rényi divergence*, given as:

**Definition 4** (Rényi divergence). For two probability distributions $P$ and $Q$ the Rényi divergence of order $\alpha > 1$ is

$$\mathsf{D}_\alpha \left( P \, \| \, Q \right) \triangleq \frac{1}{1-\alpha} \log \mathbb{E}_{x \sim Q} \left( \frac{P\left( x \right)}{Q\left( x \right)} \right)^\alpha.$$

The sensitivity of a function is an important measure of DP. It refers to the maximal change the output of a function can experience between adjacent databases and determines the magnitude of the additive noise.

**Definition 5** (Global Sensitivity). Given a function $f : \mathcal{D} \to \mathcal{R}$ for $\mathcal{R}$ as the codomain of $f$ and **all** adjacent databases as defined previously $D, D' \in \mathcal{D}$ the global sensitivity with respect to the $p$-norm is:

$$\mathrm{GS}(f) = \sup_{D,D'} \| f(D) - f(D') \|_p.$$

**Definition 6** (Local Sensitivity). Given a function $f : \mathcal{D} \rightarrow \mathcal{R}$ for $\mathcal{R}$ as the codomain of $f$ and **a specific** databases $D \in \mathcal{D}$ and its adjacent databases $D' \in \mathcal{D}$ the local sensitivity is with respect to the $p$-norm is:

$$\mathrm{LS}(f, D) = \sup_{D'} \| f(D) - f(D') \|_p.$$

DP is operationalized in DL through DP-SGD (Abadi et al., 2016). In essence, DP-SGD is different from SGD as it privatizes the minibatch (called *lot* in DP) gradients by:

1. Bounding the global sensitivity of SGD by clipping the **per-sample** gradients $g_i$ to a pre-determined by the clipping norm $C$ before aggregating them across the $B$ samples contained in the lot to a clipped gradient $g_{\text{clip}}$

$$g_{\text{clip}} = \frac{1}{B} \sum_{i=0}^{B} g_i / \max\left(1, \frac{\|g_i\|_2}{C}\right) \tag{5}$$

2. Adding zero-centered Gaussian noise with a noise multiplier $\sigma$ to the minibatch gradient

$$g_{\text{priv}} = g_{\text{clip}} + \xi, \ \xi \sim \mathcal{N}(0, C^2 \sigma^2 \mathrm{I}_\mathrm{K}) \tag{6}$$

**Relation between LSI and DP** With $\alpha \rightarrow 1$, $\mathsf{D}_\alpha (P \parallel Q) = \mathsf{KL} (P \parallel Q)$, which is justified by continuity. Intuitively, LSI thus effectively measures $(1, \epsilon)$-*per instance* RDP *post hoc*, where the privacy of a single sample (instance) is expressed with respect to a fixed dataset (Wang, 2017; Yu et al., 2022). Note that the DP guarantee of DP-SGD is diven with respect to the released gradients, but since the KL (and the Rényi) divergence both satisfy the data processing inequality (equivalently, DP is closed under post-processing), the effect translates to the model parameters. To thoroughly investigate the impact of DP on the LSI and thus on the information flow of the individual samples to the trained neural network, we examine the effects of clipping and noising separately. As the additive noise in DP-SGD introduces variance in our training setup, we average the results of DP-SGD training on CIFAR-10 across five different seeds.

**Results** With an increasing clipping norm, the LSI and, thus, the contribution of individual samples to the final parameters of the neural network decreases (Figure 17). Consequently, analyzing the LSI offers empirical support for the mathematical description that clipping gradients in DP-SGD restrict the global sensitivity and, thus, the per-sample gradients, thereby bounding the maximal information a sample can contribute to the final parameters of the model.

Notably, we show that reducing the clipping threshold results in the LSI becoming more similar across samples, with the LSI across all samples becoming less variant (Figure 17). Thus, with a decreased clipping threshold, the samples in the training data more evenly contribute their information to the neural network parameters. This aligns with previous findings showing the importance of dataset variability on generalization (Therrien & Doyle, 2018) and the beneficial role of DP for generalization (Nissim & Stemmer, 2015).

Our experiments empirically show that adding zero-centered Gaussian noise scaled with respect to the maximum gradient norm of the gradients has no observable influence on the distribution of LSI in the dataset (Figure 17). While the clipping of individual gradients only influences those samples whose gradients are larger than the clipping threshold, thereby changing the amount of information individual samples contribute to the training, the noise gets added to all samples. The noise increases the variance in the per-sample gradients. However, their average contribution to the neural network parameters remains unchanged. This aligns with the findings of previous research, showing adding noise solely influences the privacy risks of training the model, but not the convergence or calibration, while the gradient clipping solely influences the convergence and calibration of the neural network (Bu et al., 2021).

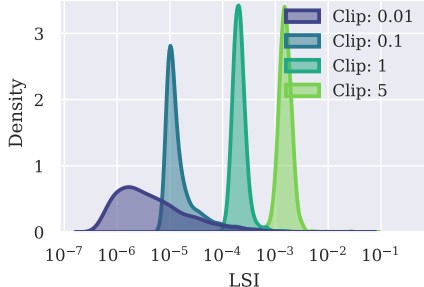 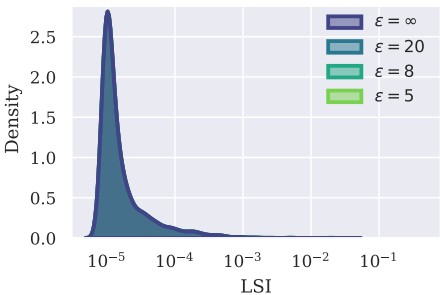

Figure 17: The left plot shows the distribution of the LSI of CIFAR-10 across varying per-sample gradient clipping norms with no additive noise. The right shows the distribution of the LSI with clipping norm $0.1$ and varying additive noise to achieve a $(\epsilon, 10^{-5})$-DP. With diminishing clipping norm, LSI decreases in magnitude and variance, while with increasing $\epsilon$, the distribution of LSI remains unchanged in magnitude and variance.

# I  GENERALIZATION TO RIEMANN LAPLACE APPROXIMATION

Recently, a novel approximation of the posterior model distribution has been proposed in Bergamin et al. (2024), which allows for sampling models while considering the true shape of the underlying loss landscape. While computing the LSI on this Riemannian Laplace approximation (RLA) is possible, it requires fitting a distribution via Kernel Density Estimation to the samples produced by the RLA. These distributions are then used to compute the LSI. We show, using the same sampling and fitting approach on the standard Laplace approximation, that while this process produces the correct LSI values for "intermediately" spaced parameter distributions (Figure 18), it breaks down when the distributions are very similar or very distant (Figure 19). Very similar distributions (which is the setting in leave-one-out retraining) require many samples to be drawn to capture the slight differences in distributions. Similarly, very distant distributions require inordinately many samples to capture the similarities in distributions. RLA additionally requires solving a differential equation for each sample of model parameters. Generating sufficiently many samples to adequately describe the distributions is thus computationally infeasible, especially as the sampling would have to be performed each retraining cycle. Due to these reasons, we use the "traditional" Laplace approximation in our work.

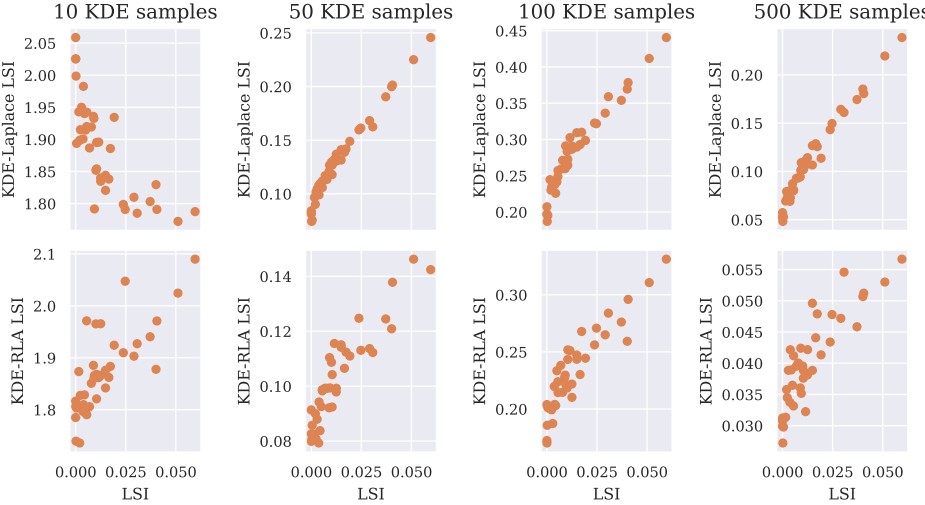

Figure 18: Correlation between LSI computed on the kernel density estimate drawn from Riemann Laplace approximation, the kernel density estimate drawn from Laplace approximation and LSI computed on the gaussian distributions estimated by the Laplace approximation without kernel density estimation. The figure shows LSI computed on distributions with **intermediate LOO distribution shift**.

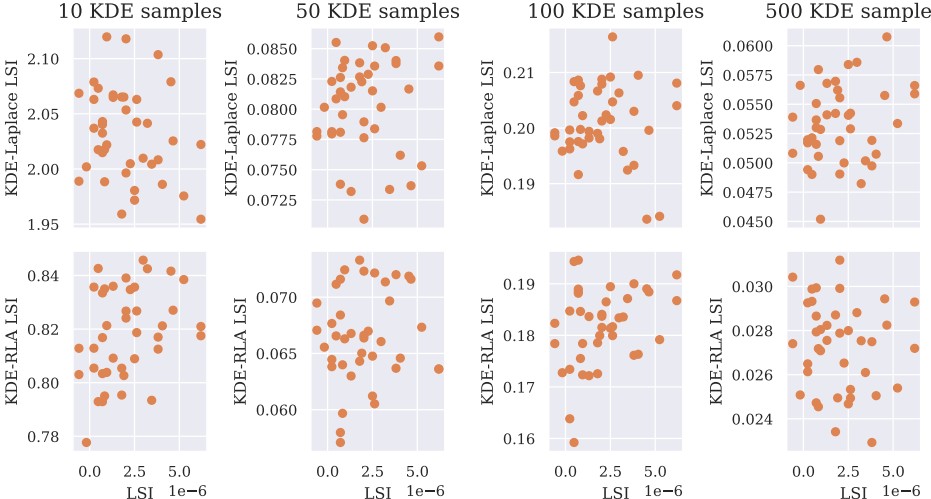

Figure 19: Correlation between LSI computed on the kernel density estimate drawn from Riemann Laplace approximation, the kernel density estimate drawn from Laplace approximation and LSI computed on the gaussian distributions estimated by the Laplace approximation without kernel density estimation. The figure shows LSI computed on distributions with **small LOO distribution shift**.

## J    ADDITIONAL RESULTS ON TRANSFERRING LSI FROM LINEAR PROBE TO FULL MODEL

In this section, we provide additional results on transferring the LSI ordering computed on the probe (called proxy in the figures) to other architectures. We are considering the following models: the probe, an MLP, a three-layer CNN, a ResNet-9 and a ResNet-18 for CIFAR10 and CIFAR100, and a ResNet-18 for the ImageNet subsets and the *pneumonia* dataset.

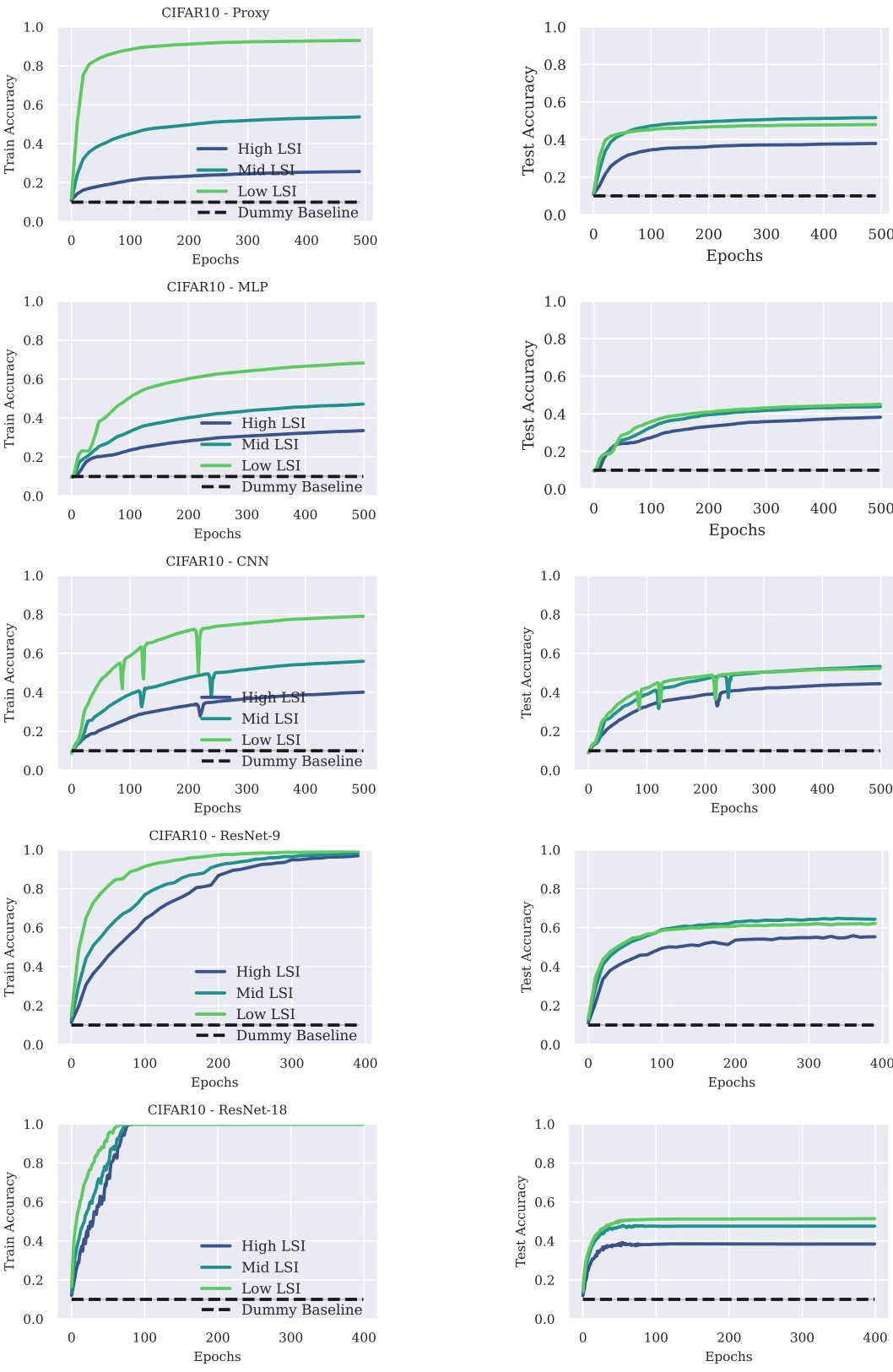

Figure 20: Accuracy of LSI based subsets of CIFAR10 across an MLP (flattened input), CNN, ResNet-9, ResNet-18

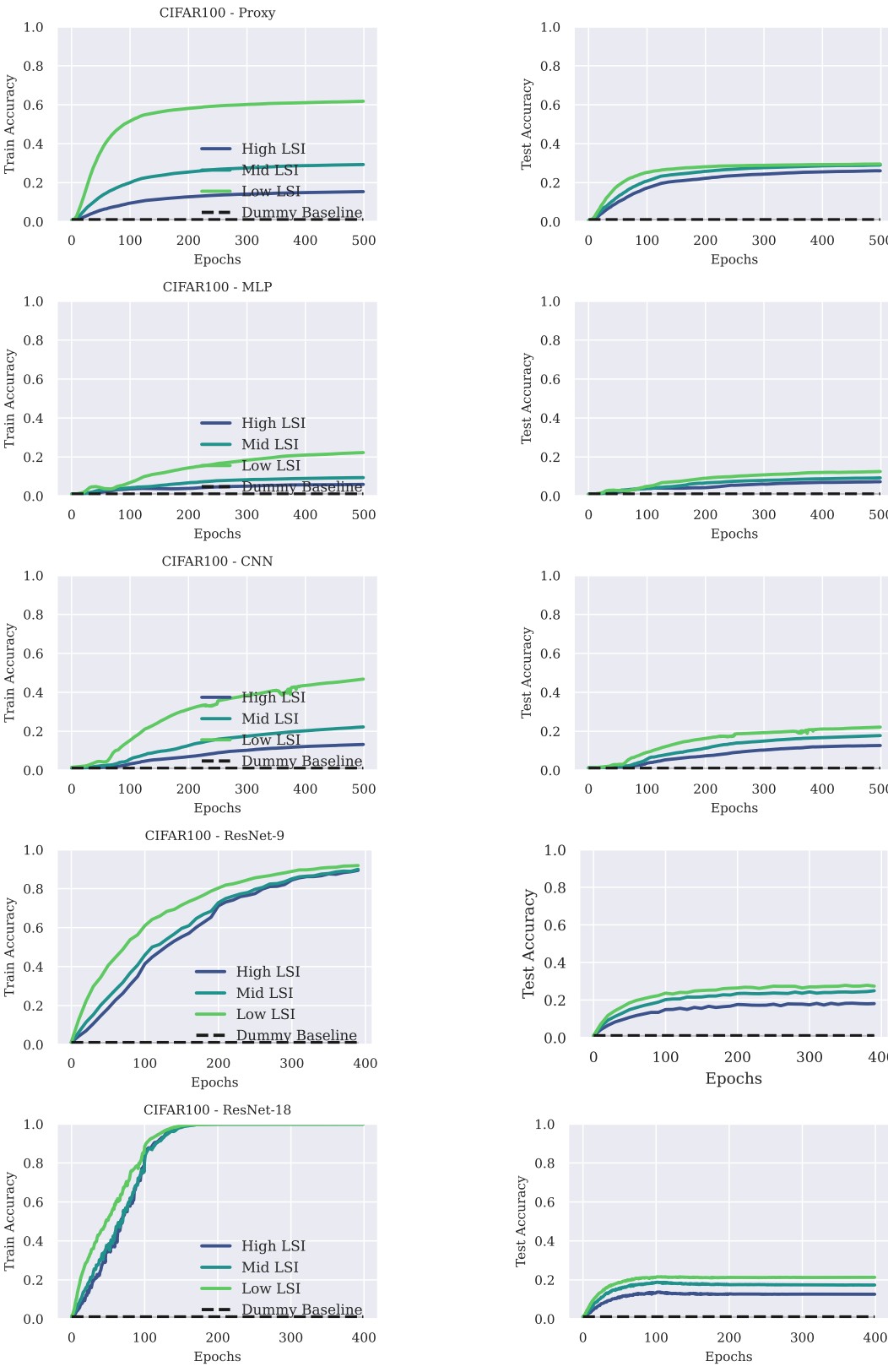

Figure 21: Accuracy of LSI based subsets of CIFAR100 across an MLP (flattened input), CNN, ResNet-9, ResNet-18

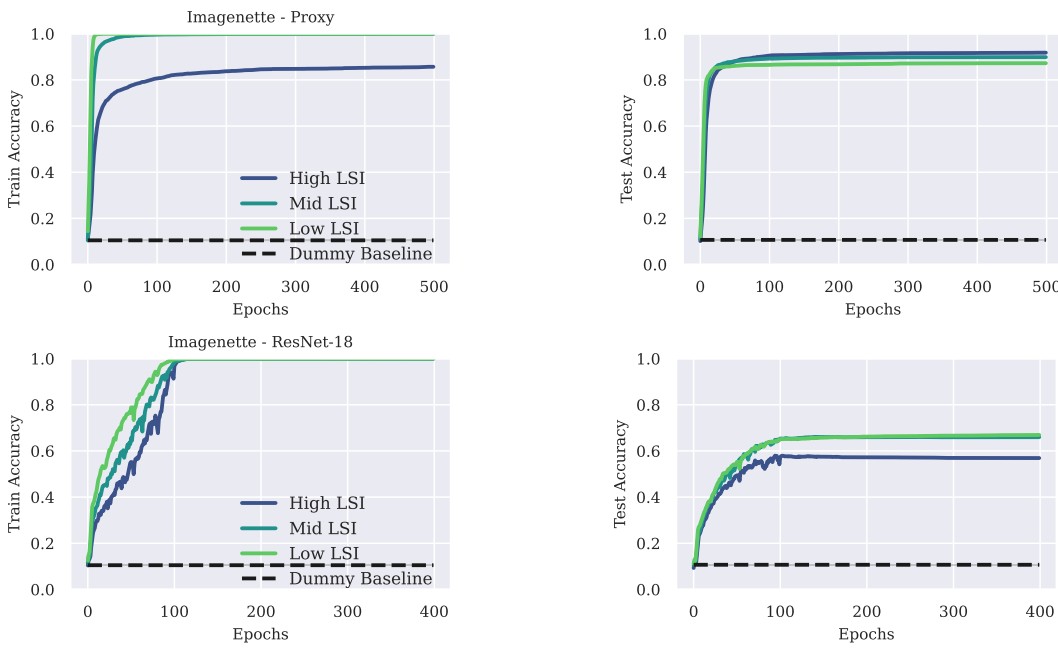

Figure 22: Accuracy of LSI based subsets of Imagenette across ResNet-18

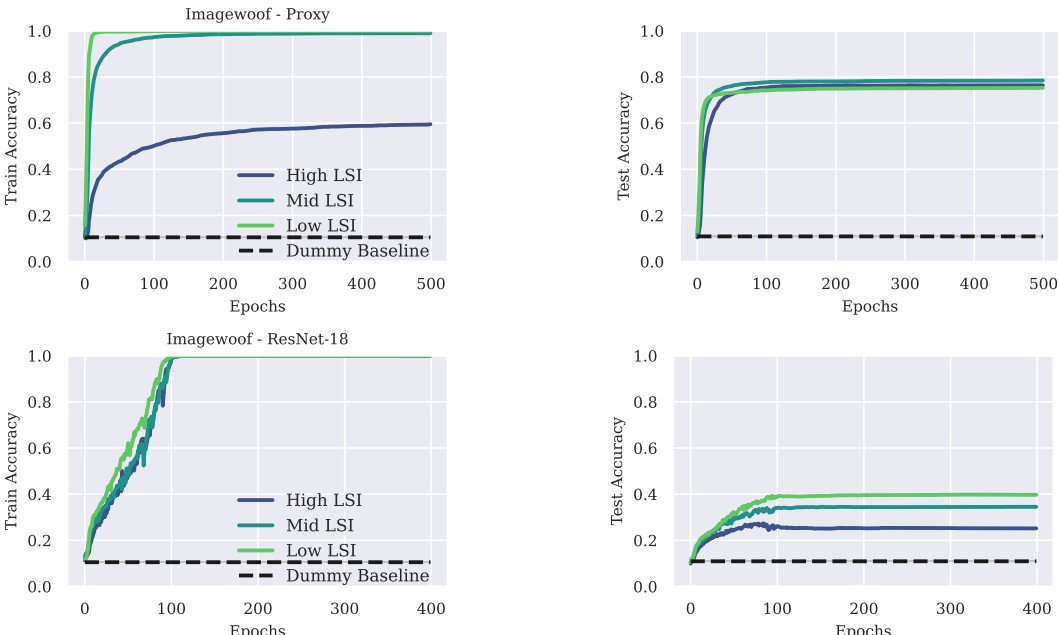

Figure 23: Accuracy of LSI based subsets of Imagewoof across ResNet-18

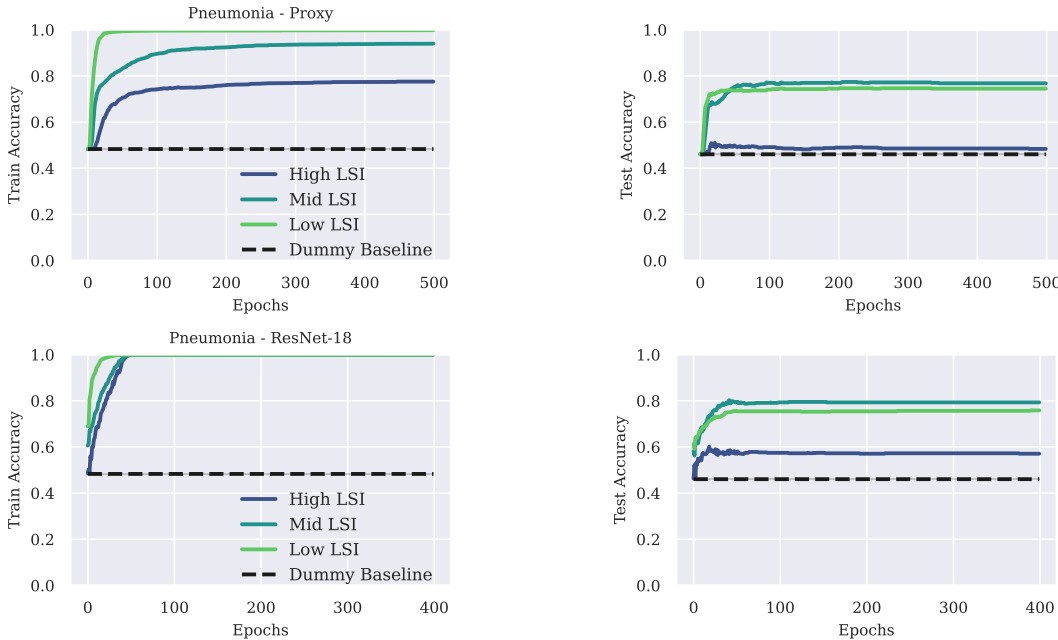

Figure 24: Accuracy of LSI based subsets of the Pneumonia dataset across ResNet-18

## K    EXEMPLARY IMAGES OF IMAGENET WITH LOW AND HIGH **LSI**

We show the applicability of LSI on ImageNet (Deng et al., 2009) using two popular subsets, Imagenette and Imagewoof (Howard, 2019). While Imagenette contains a subset of easily distinguishable classes, Imagewoof solely contains classes of dogs and is considered more difficult. Computing the LSI allows the detection of mislabeled examples (e.g. the image of the car in the class *radio*). The following figures show samples of high and low LSI, with mislabeled examples indicated by a red dot (to the best of our knowledge) and with a yellow dot if an instance of another class is present in the image.

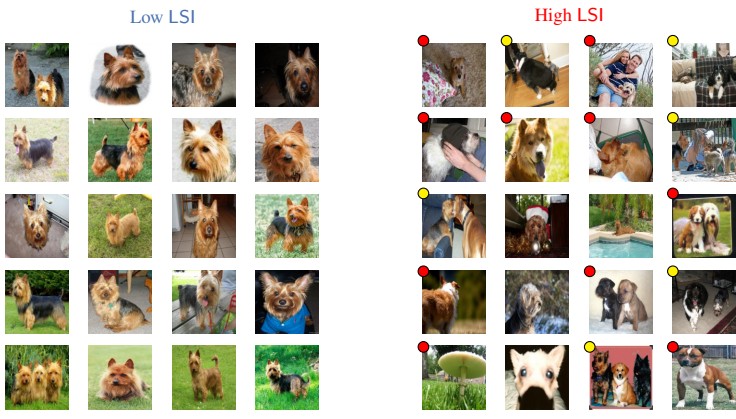

Figure 25: Exemplary images of the highest and lowest LSI of the *Australian terrier* class. Note the presence of images of dogs with different breeds and images of non-dogs (e.g. the mushroom) in the high LSI parition.

Low LSI                    High LSI

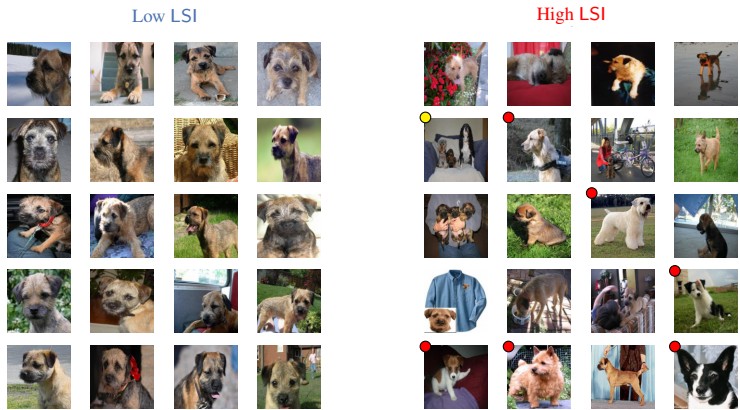

Figure 26: Exemplary images of the highest and lowest LSI of the *Border terrier* class. Note again the presence of different breeds and atypical images (e.g. the shirt with a small dog logo) in the high LSI parition.

Low LSI                    High LSI

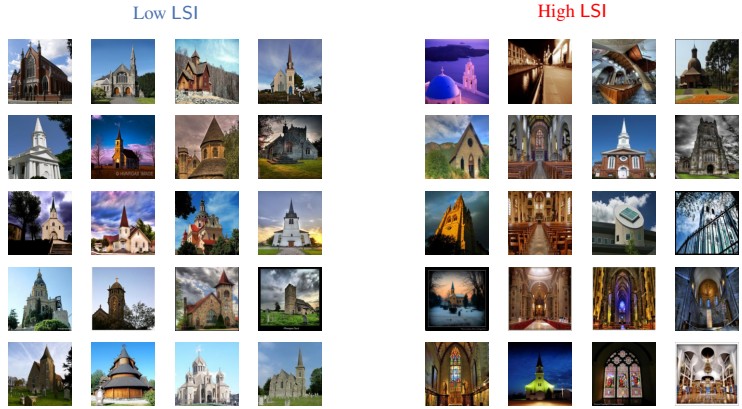

Figure 27: Exemplary images of the highest and lowest LSI of the *church* class. Note that low LSI images mostly show outside views of churches, while high LSI images show churches from uncommon perspectives and indoor views, which could be confused with other indoor spaces).

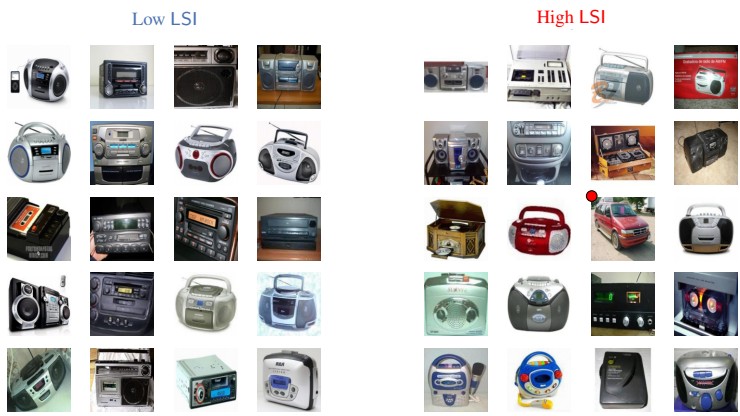

Figure 28: Exemplary images of the highest and lowest LSI of the *radio* class. Note that the the high LSI partition includes several uncommon-looking radios (e.g. the children's radio in the bottom row or the antique radio in the third row, and at least one image of a different class (the red car).

## L  EXEMPLARY IMAGES OF CIFAR-10 WITH LOW AND HIGH LSI

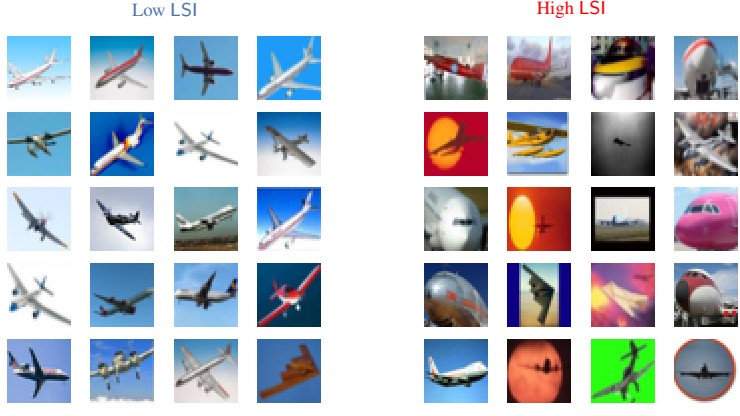

Figure 29: Exemplary images of the highest and lowest LSI of the *airplane* class. Note the differences in perspective and background color.

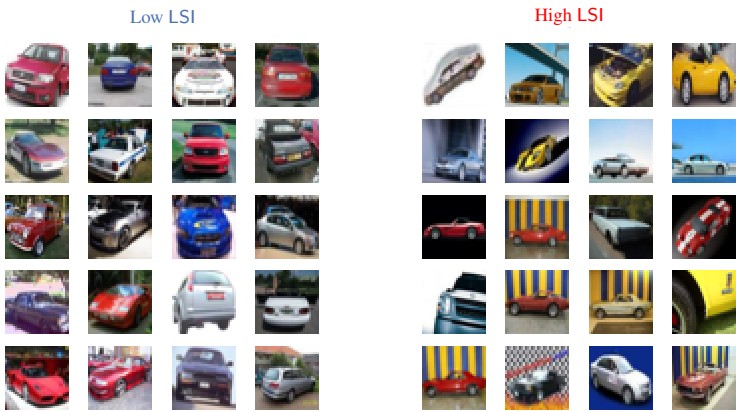

Figure 30: Exemplary images of the highest and lowest LSI of the *automobile* class. Note the differences in background and subject size in the frame.

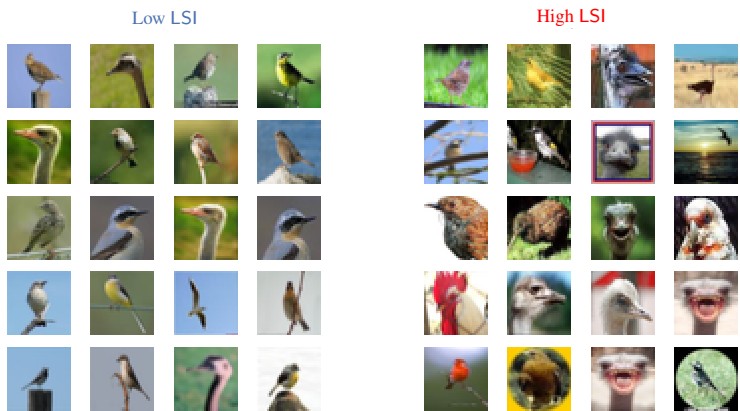

Figure 31: Exemplary images of the highest and lowest LSI of the *bird* class. Note the overrepresentation of close-crop ("portrait") images of bird heads in the high LSI partition compared to the mostly whole-bird images in the low LSI partition.

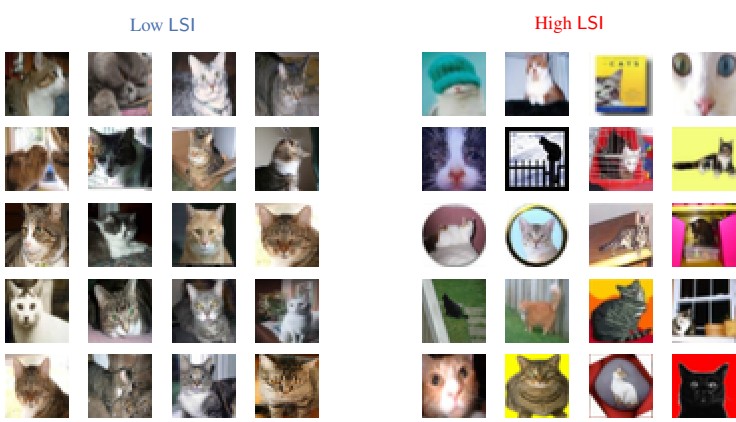

Figure 32: Exemplary images of the highest and lowest LSI of the *cat* class. Note the heterogeneity of the high LSI partition, including atypical backgrounds and different cropping.

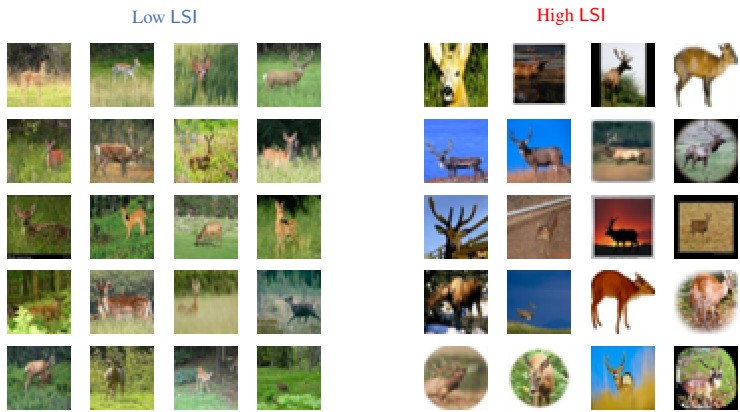

Figure 33: Exemplary images of the highest and lowest LSI of the *deer* class. Note the heterogeneity of the high LSI partition, including atypical backgrounds and different cropping.

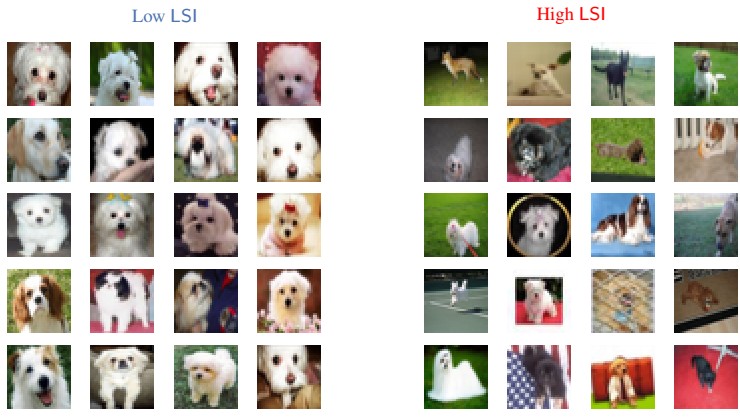

Figure 34: Exemplary images of the highest and lowest LSI of the *dog* class. Note the presence of at least one non-dog (the fox in the top row) and the inconsistent framing and subject size in the high LSI parition compared to the high homogeneity of the low LSI parition.

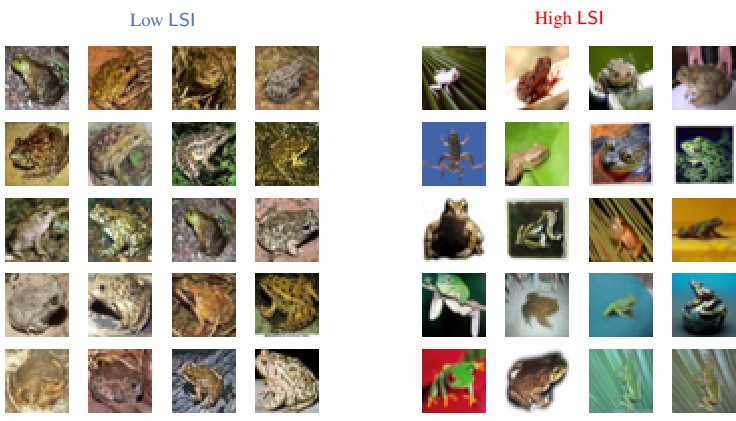

Figure 35: Exemplary images of the highest and lowest LSI of the *frog* class. Note the heterogeneity of the high LSI partition, including atypical backgrounds and different cropping, the presence of an albino frog and at least two images where the subject is camouflaged (bottom row).

Low LSI         High LSI

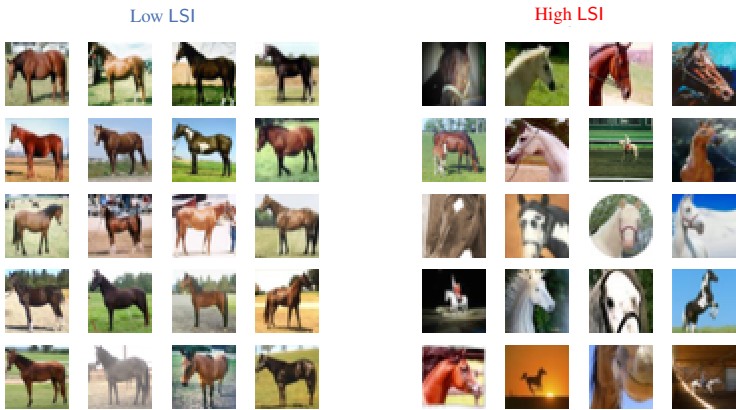

Figure 36: Exemplary images of the highest and lowest LSI of the *horse* class. Note the heterogeneity of the high LSI partition, including different framing, cropping and backgrounds, as well as white horses, whereas the low LSI partition contains mostly brown horses.

Low LSI         High LSI

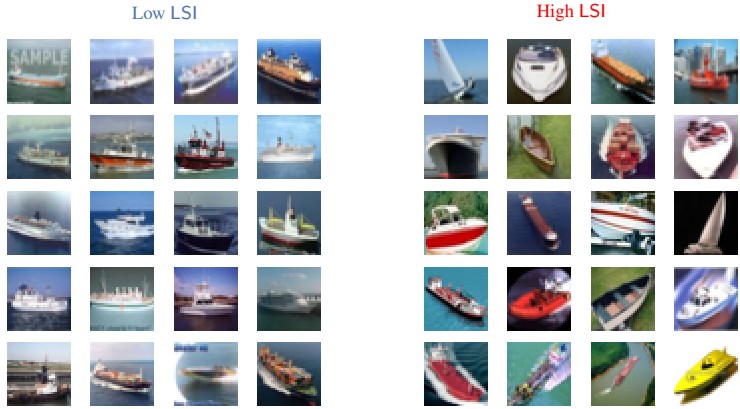

Figure 37: Exemplary images of the highest and lowest LSI of the *ship* class. Note the heterogeneity of the high LSI partition and the different backgrounds compared to the standard backgrounds in the low LSI partition.

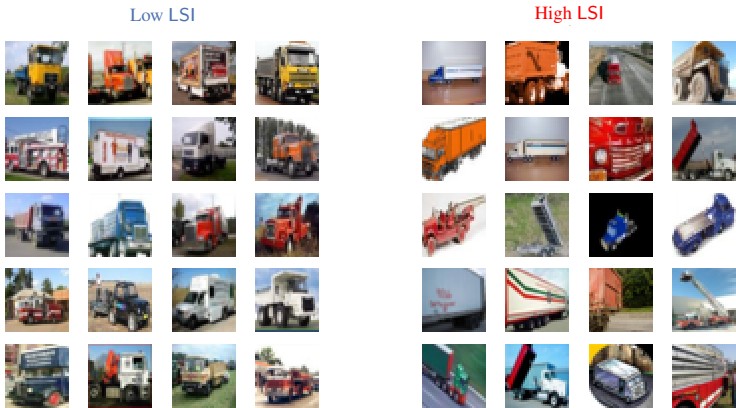

Figure 38: Exemplary images of the highest and lowest LSI of the *truck* class. Note the atypical perspectives in the high LSI partition, as well as the inclusion of at least one ambiguous image (the trailer in the third row).

## M  EXEMPLARY IMAGES OF MEDICAL IMAGING DATA (PNEUMONIA) WITH LOW AND HIGH LSI

Examples carrying low LSI are similar to each other and are captured with a very symmetrical view. Samples carrying high LSI are angled, have a different exposure, or contain samples that are not supposed to be in the dataset (images of adult females in a dataset of child pneumonia x-ray images).

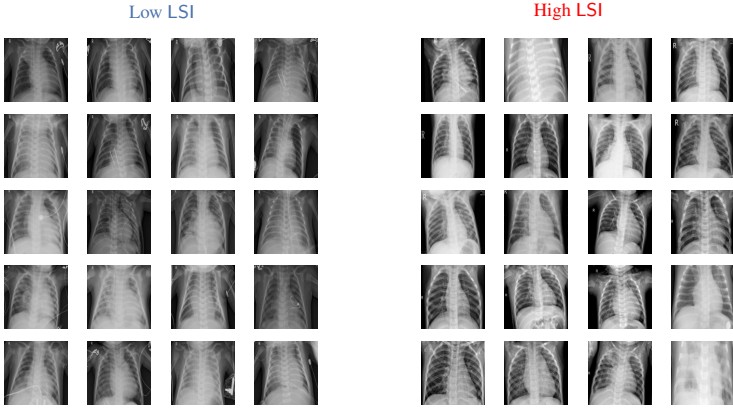

Figure 39: Exemplary images of the highest and lowest LSI of the *bacterial* class.

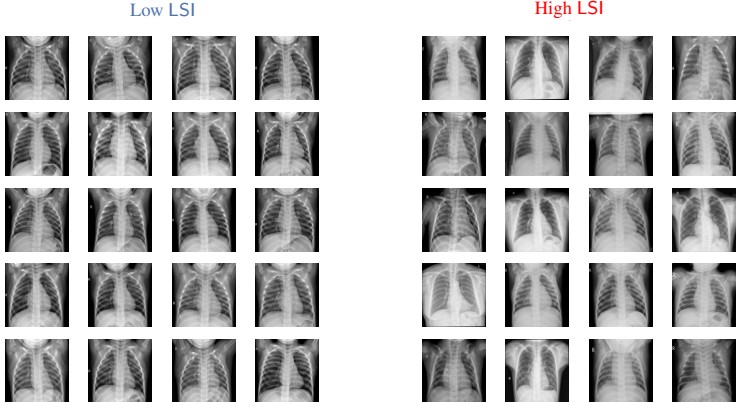

Figure 40: Exemplary images of the highest and lowest LSI of the *normal* class.

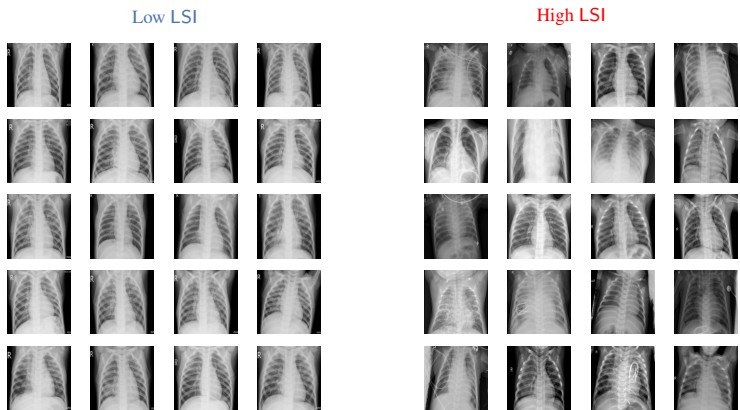

Figure 41: Exemplary images of the highest and lowest LSI of the *viral* class.

## N  TRAINING PARAMETERS FOR THE EXPERIMENTS

Table 2: Training parameters for each of the experiments

| Parameter | LSI-Distribution | Sample Difficulty/ Generalization | LSI under Differential Privacy |
|---|---|---|---|
| Learning rate | 0.04 | 0.04 | 0.04 |
| Weight decay (L2) | 0.01 | 0.01 | 0.01 |
| Nesterov Momentum | 0.9 | 0.9 | 0.9 |
| Dataset | Full | 1/3 Subsets | 1/5 Subsets |
| Epochs/ Steps | 1000 | 1000 | 700 |
| Averaged across n Seeds | 3 | 3 | 5 |

## O  HARDWARE SETUP AND COMPUTATIONAL RESSOURCES

All experiments described in this paper and its appendix are performed on an *NVidia H100* GPU (80GB VRAM) with 2 *AMD EPYC 9354 32-Core* CPUs. While we employ an 80GB GPU, the required VRAM is far smaller, such that LSI can easily be computed on GPUs of solely 24GB of VRAM. Table 3 shows the time it took to compute the LSI for each sample in each dataset used in

this publication. Note, that the retraining requires the repeated transfer of data from the CPU Memory to the GPU Memory. Thus, computing the LSI on the GPU rather than the CPU provides solely a speedup of around 2, which potentially provides room for improvement.

Table 3: Training parameters for each of the experiments

|  | CIFAR-10 | CIFAR-100 | Pneumonia | Imagenette | Imagewoof |
|---|---|---|---|---|---|
| Compute (h) | ~18 | ~18 | ~0.5 | ~1.7 | ~1.7 |

The computation of LSI under differential privacy was performed on subsets of 10000 samples on CIFAR-10 and required ~12h of compute. All other experiments required <1h of compute. Preliminary experiments required a total compute of about $5\times$ the computational time that yielded the results that are shown in this manuscript.

## P    FUNDING ACKNOWLEDGEMENTS

GK received support from the German Ministry of Education and Research and the Medical Informatics Initiative as part of the PrivateAIM Project, from the Bavarian Collaborative Research Project PRIPREKI of the Free State of Bavaria Funding Programme *Artificial Intelligence – Data Science*. This work was partially funded by the Konrad Zuse School of Excellence in Reliable AI (relAI).

This work is co-funded by the European Union under Grant Agreement 101100633 (EUCAIM). Views and opinions expressed are however those of the author(s) only and do not necessarily reflect those of the European Union or the European Commission. Neither the European Union nor the granting authority can be held responsible for them.

