# OpenReview forum: "Laplace Sample Information:  Data Informativeness Through a Bayesian Lens"
_ICLR.cc/2025/Conference — ICLR 2025 Poster_

### Official Review · Reviewer_qsLf · 2024-10-31

**Soundness:** 3
**Presentation:** 3
**Contribution:** 3
**Rating:** 6
**Confidence:** 4

**Summary:**

This paper considered an information theoretical framework in understanding the data sample importance. Specifically, the intuition is further formulated as the difference of parameter distributions. Experiments on various and extensive experiments demonstrate the effectiveness of the proposed methods.

**Strengths:**

- I think this paper raised an important problem in explaining the important data sample.
- The proposed method is theoretically principled with Bayes principle.
- Extensive experimental results are presented with CV and NLP datasets (since this method is model agnostic).
- The paper is clearly written and accessible.

**Weaknesses:**

There are two major weak points within the paper. Therefore I currently provide a borderline score.

1. About the novelty. How this method is different from the influence function based method. Both methods require some calculation on the higher-order derivative. Based on the given context, I find it a bit hard to differentiate the differences.
2. About the calculation LSI in line 212, I could not understand how exactly \hat_{theta}^{-i} and \Sigma^{-i} are estimated. Here the technical detail is very limited, therefore I could not understand why estimating these can be done after training. Please give clear details how LSI is calculated.
3. (Minor) The paper is a bit large by itself, 32Mb ! I could not properly read the results due to its large file.

**Questions:**

See the questions in weakness sections.

**Details Of Ethics Concerns:**

I would like author to discuss its potential negative implications of this methods such as data leakage issue.

---

> ### Author Response · Authors · 2024-11-19
> **Author Comment on the Weaknesses described by Reviewer qsLf**
>
> Thank you for reviewing our work and pointing out strengths and weaknesses.
>
> ---
>
> > About the novelty. How this method is different from the influence function based method. Both methods require some calculation on the higher-order derivative. Based on the given context, I find it a bit hard to differentiate the differences.
>
> Thank you for the comment. The conceptual difference between our method and Influence functions is substantial: Influence functions approximate the influence of an individual sample by solely considering the loss of a sample, as well as the Hessian at the location of convergence of the model. However, the presence of a single data point can influence the entire training trajectory of the model, yielding a completely different set of converged weights. Thus, estimating the influence by starting from a converged model as done by influence functions is merely an *extrapolation of the true counterfactual influence* of the sample (in fact, influence functions are known to be fragile [1, 2, 3]). LSI is based on leave-one-out retraining, which estimates the *actual influence* of a data point and is thus a full data attribution method [4]. However, as noted in the discussion of the manuscript, this comes at the cost of additional time complexity. However, seeing as it is parallelizable, LSI remains computationally tractable even for larger datasets.
>
> ---
>
> > About the calculation LSI in line 212, I could not understand how exactly $\hat{\theta}^{-i}$ and $\Sigma^{-i}$ are estimated. Here the technical detail is very limited, therefore I could not understand why estimating these can be done after training. Please give clear details how LSI is calculated.
>
>  $\hat{\theta}^{-i}$ and $\Sigma^{-i}$  are described in lines 206 and 208 of the original manuscript and constitute the final weights of the trained model with the left out datapoint and the curvature of the loss landscape at the final weights respectively. While the first is approximated with the probe model, the latter is computed using automatic differentiation given the final weights.
>
> ---
>
> > (Minor) The paper is a bit large by itself, 32Mb ! I could not properly read the results due to its large file.
>
> Thank you for your feedback regarding the file size. We understand that the large file size may have affected readability primarily due to high-resolution figures and a lengthy appendix. To address this, we plan to replace some SVG files with high-resolution PNGs, reducing the overall file size and improving accessibility for readers.
>
> ---
>
> > I would like author to discuss its potential negative implications of this methods such as data leakage issue.
>
> Thank you for this comment. Unfortunately, we are uncertain what you mean by “data leakage issue”.  Would you please clarify so that we can address any concerns meaningfully.
>
> If you are referring to data points influencing each other, since we do real LOO retraining, we expect no data leakage between the points.
>
> In case you mean data privacy concerns, refer to Appendix F, where we include a discussion and investigation of the parallels between LSI and Differential Privacy, a standard algorithmic privacy technique, which we also intend to explore in future work.
>
> ---
>
> [1] Samyadeep Basu, Phillip Pope, and Soheil Feizi. “Influence Functions in Deep Learning Are Fragile”. In: International Conference on Learning Representations (ICLR). 2021
>
> [2] Andrew Ilyas, Sung Min Park, Logan Engstrom, Guillaume Leclerc, and Aleksander Madry. “Datamodels: Predicting Predictions from Training Data”. In: International Conference on Machine Learning (ICML). 2022
>
> [3] Ekin Akyurek, Tolga Bolukbasi, Frederick Liu, Binbin Xiong, Ian Tenney, Jacob Andreas, and Kelvin Guu. “Towards Tracing Factual Knowledge in Language Models Back to the Training Data”. In: Findings of EMNLP. 2022.
>
> [4] Cook, R. Dennis; Weisberg, Sanford. (1982). Residuals and Influence in Regression.

---

> > ### Author Response · Authors · 2024-11-26
> > **Revised Version of the Submission**
> >
> > Dear Reviewer qsLf,
> >
> > We have just now uploaded a revised version of the document, which is now about 30% smaller in size, with larger figures limited to the appendix, which should aid with readability.
> >
> > Furthermore, we have conducted additional experiments comparing LSI with TRAK (Appendix H) and on the effect of human mislabeled data on LSI (Appendix I).
> >
> > We hope this sufficiently addresses all your points. If not, we would be happy to engage in further discussion.

---

> > > ### Comment · Reviewer_qsLf · 2024-12-02
> > > **Acknowledgement**
> > >
> > > I would appreciate your response. After reading other reviews and rebuttal. I maintained my current score.

---

> ### Author Response · Authors · 2024-11-25
> **Feedback on Rebuttal Response**
>
> Dear Reviewer qsLf,
>
> We appreciate your input. Having carefully addressed your comments, we hope that it clarifies the points you raised. Could you confirm if our response has clarified your concerns, or if there are additional aspects that you feel still need to be addressed? We are happy to engage in a discussion during the remaining rebuttal period.

---

### Official Review · Reviewer_1zef · 2024-11-03

**Soundness:** 3
**Presentation:** 3
**Contribution:** 3
**Rating:** 6
**Confidence:** 4

**Summary:**

The paper describes a novel approach to measuring the informativeness of individual samples in machine learning datasets, named the Laplace Sample Information (LSI). LSI is designed to quantify how much information each sample contributes to the model's parameters using a Bayesian approximation. The method applies the Laplace approximation to generate a distribution over model parameters and subsequently employs Kullback-Leibler divergence to measure each sample's unique contribution to this distribution.

Experiments on CIFAR-10, CIFAR-100, ImageNet subsets (Imagewoof and Imagenette), a pediatric pneumonia dataset, and the IMDb text sentiment dataset. LSI revealed that only a small subset of samples contributes significant information to the model, while most samples have low informativeness. On CIFAR-10 with 10% label noise, mislabeled samples generally had higher LSI values than correctly labeled samples, showing that LSI can effectively highlight mislabeled data. Other experiments collectively validated LSI’s utility across various applications: ranking samples by informativeness, detecting mislabeled data, assessing dataset and class difficulty, and improving model generalization.

**Strengths:**

- LSI offers a new perspective on sample informativeness, focusing directly on sample information flow rather than related metrics like sample difficulty. This approach addresses a gap in the current literature, where sample informativeness has often been approximated indirectly.

- LSI is adaptable across different model architectures, learning tasks (supervised and unsupervised), and data modalities (e.g., images, text). LSI has valuable applications in identifying mislabeled samples, estimating dataset difficulty, and detecting out-of-distribution data.

- By using small probe models, the authors make LSI computationally feasible for larger models and datasets. This approach yields a substantial efficiency gain without significantly compromising the integrity of the informativeness rankings.

- The authors evaluate LSI on various datasets, including CIFAR-10, CIFAR-100, ImageNet subsets, and text datasets, with extensive experiments validating LSI’s effectiveness in distinguishing sample informativeness. They also investigate LSI’s behavior under label noise.

**Weaknesses:**

- The experiments mainly evaluate the proposed LSI measures. How does the proposed measure compare to other existing related measures, such as gradient similarity, influence functions, and TRAK (Park et al., ICML'23)?

- Although LSI uses a probe model for efficiency, computing LSI still requires repeated training sessions, especially in large datasets. It would be better to describe its computation complexity and compare it with existing measures.

- The reliance on the Laplace approximation assumes that the model parameters follow a multivariate Gaussian distribution. While the authors justify this through the central limit theorem, it would be better to provide empirical justifications, especially for highly complex or non-convex loss landscapes.

- The method relies on approximations of the Hessian, particularly when using diagonal approximations for large models. It would be better to analyze how this approximation affects LSI in different neural architectures.

**Questions:**

Please see weaknessnes

---

> ### Author Response · Authors · 2024-11-19
> **Author Comment on the Weaknesses described by Reviewer 1zef**
>
> > Compare LSI to Influence and TRAK
>
> Thank you for the suggestion. We already include a comparison to other existing methods in Appendix A, namely for pointwise sliced mutual information and smooth unique information. We opted for these metrics, as they are based on estimating the mutual information of a sample point and the weights and are thus closest to LSI. The values or our method correlate well with both of these, while pointwise sliced mutual information and smooth, unique information are uncorrelated. This indicates that these two methods measure distinct but related quantities, while LSI measures both.
>
> Influence functions estimate the influence of an individual sample by solely considering the loss of a sample, as well as the Hessian at the location of convergence of the model. However, the presence of a single data point can influence the whole training trajectory of the model, yielding a completely different set of converged weights. Thus, estimating the influence by starting from a converged model as done by influence functions is a hypothetical extrapolation of the true influence (in fact, influence functions are known to be fragile [1, 2, 3]). Thus, these two measures estimate two different things.
>
> However, as Influence functions are widely used in practice, we agree that comparing LSI with Influence functions is beneficial for contextualizing our work. Therefore we will include the comparison in the camera-ready version of the manuscript by extending Appendix A.
>
> Our comparison shows that, while LSI is moderately correlated with TRAK as well as influence scores, there are distinct samples that carry high unique information (as measured by LSI) but have low influence or Trak scores. Further investigation revealed that a subset of these strongly uncorrelated examples are mislabeled or strongly out-of-distribution samples. This indicates that (1) while Influence functions and TRAK measure quantities that are related to the unique information(2) LSI allows for detecting out-of-distribution samples which both influence functions and TRAK miss. This is in line with the known fragility of Influence functions and TRAK and supports the use of our method.
>
> ---
>
> > Comparison of the computational complexity of LSI
>
> Thank you for the thoughtful comment. You are raising a valid point. However, comparing LSI with existing methods is more complex than solely comparing based on the computational complexity. In fact, all data-attribution methods are trading off computational complexity, accuracy, and scalability. Therefore, it is very much dependent on the circumstances in which data attribution method is to be preferred, as none is objectively better than any other. LSI is linear in time O(kN) (k as the number of training steps and N as the number of samples) and constant O(N) (with N as the number of parameters of the probe) in space when using our proposed diagonal Hessian approximation. LSI is, therefore very efficient w.r.t. space complexity, it is well-founded as an approximation of the pointwise conditional mutual information and shown to be robust. Additionally, the space requirements of LSI remain constant w.r.t. the sample count.
>
> Comparably, Influence Functions and their approximations at worst require the computation of the Hessian, which is O(N^2) in time and space, and also its inversion, which is O(N^3) in time and most importantly, are additionally known to be fragile [1, 2, 3] as they estimate the influence of a sample based on a one Newton step from the converged model. Recently proposed TRAK [4] speeds up influence functions by reducing them to logistic regression, which consequently shares the same fragility problems of influence functions and comes at the cost of introducing additional approximation errors. Smooth unique Information [5] requires space that scales with the dimensionality of the dataset and is thus not applicable to larger datasets (> 1000 samples). [6, 7] requires the training of hundreds of subsampled models until convergence to estimate the influence of samples.
>
> Therefore, while it is possible to compare our numerical compare our method with other measures, it would not provide a full picture of the complicated task of data attribution and its inherent trade-offs.
>
> ---
>
> [1] Samyadeep Basu et al. “Influence Functions in Deep Learning Are Fragile” (2021)
>
> [2] Andrew Ilyas et al. “Datamodels: Predicting Predictions from Training Data” (2022)
>
> [3] Ekin Akyurek et al. “Towards Tracing Factual Knowledge in Language Models Back to the Training Data”. (2022)
>
> [4] Park, Sung Min, et al. "Trak: Attributing model behavior at scale." (2023).
>
> [5] Harutyunyan, Hrayr, et al. "Estimating informativeness of samples with smooth unique information." (2021).
>
> [6] Vitaly Feldman and Chiyuan Zhang. “What Neural Networks Memorize and Why: Discovering the Long Tail via Influence Estimation” (2020)
>
> [7] Ilyas, Andrew, et al. "Datamodels: Predicting predictions from training data." (2022).

---

> ### Author Response · Authors · 2024-11-19
> **Author Comment on the Weaknesses described by Reviewer 1zef continued**
>
> > The reliance on the Laplace approximation assumes that the model parameters follow a multivariate Gaussian distribution. While the authors justify this through the central limit theorem, it would be better to provide empirical justifications, especially for highly complex or non-convex loss landscapes.
>
> Thank you for the comment, you are raising a valid point, and we had the same concern. That is why we conducted experiments investigating the applicability of LSI on neural networks prior to their convergence, which you can find in Appendix E of the original manuscript. Our results show that after a brief warm-up period of a few epochs, the order of samples established via LSI does not substantially change. This indicates that even though LSI formally requires the convergence of the model, it remains applicable far prior to convergence.
>
> Moreover, while our experiments consider a Gaussian posterior distribution of the weights, our method can be extended to other (more sophisticated) posterior approximation methods. An investigation thereof is included in Appendix G of the original manuscript, where we extend LSI to the recently proposed Riemann Laplace approximation.
>
> ---
>
> > The method relies on approximations of the Hessian, particularly when using diagonal approximations for large models. It would be better to analyze how this approximation affects LSI in different neural architectures.
>
> Thank you for this comment, which is a valid concern we also shared. We would like to refer you to Appendix B, comparing the sample ordering established with LSI computed with different approximations of the Hessian. Our experiments show that LSI computed on the diagonal approximation correlates excellently with LSI computed on the full Hessian (Spearman’s Rank Correlation Coefficient of 0.93). This aligns with concurrent research showing that the Hessian of later layers of neural networks is diagonal dominant [1]. Appendix B also includes the same analysis with respect to the KFAC representation of the Hessian.
>
> Furthermore, Appendix C of the original manuscript compares the sample ordering established with LSI computed on the probe model vs. the sample ordering of LSI computed on a CNN. With excellent correlation (also Spearman’s Rank Correlation Coefficient of 0.93) our results indicate that the LSI computed on the probe model is a suitable approximation.
>
> ---
>
> [1] Elsayed, Mohamed, et al. "Revisiting Scalable Hessian Diagonal Approximations for Applications in Reinforcement Learning." (2024).

---

> > ### Comment · Reviewer_1zef · 2024-11-24
> >
> > Thanks for the authors' responses!
> >
> > The authors show the comparison between the proposed LSI and pointwise sliced mutual information and smooth unique information. Yet, I think it is better to provide a more empirical comparison with existing data attribution methods. Moreover, I think these comparisons should be discussed in the main content of the paper.
> >
> > Also, the authors claim that "these two measures estimate two different things" (between influence functions). I still think at a high-level, the two measures are targeted for a similar goal.
> >
> > The rest of my concerns have been addressed. Thus, I would like to increase my score accordingly.

---

> ### Author Response · Authors · 2024-11-26
> **Revised Version of the Submission**
>
> Thank you for raising the score and the additional feedback.
>
> We have now uploaded a revised version of the original submission, including the comparison of LSI with TRAK in **appendix H**, which also includes a short remark on the difference of influence estimation (as done by TRAK) and data attribution with LSI. We agree that the main body of the work would benefit from an empirical evaluation. However, as there is no ground truth in data valuation, this often requires extensive discussion and, therefore, space. Consequently, we decided to move the comparison to the beginning of the appendix to allow the main body to focus on our contribution while making the empirical comparisons easily accessible and deliberately referring to it in the main body.
>
> *(Note, that we chose to append the comparison of LSI and TRAK here to not mess with references on appendices of the original manuscript we made in the rebuttel and will reposition it as Appendix B in the camera ready version of the work.)*
>
> Moreover, the revised Version of the document includes an additional experiment on human mislabeled data in Appendix I as well as it being smaller in size (in terms of memory) to aid with readability.
>
> If you have any remaining concerns, please let us know so we can address them.

---

### Official Review · Reviewer_6Bs2 · 2024-11-04

**Soundness:** 2
**Presentation:** 3
**Contribution:** 3
**Rating:** 6
**Confidence:** 3

**Summary:**

This paper presents Laplace Sample Information (LSI), a metric to quantify how much individual data points contribute to a neural network’s learning process. By approximating a Bayesian model through the Laplace method, LSI uses KL divergence to capture each sample’s impact on the model’s parameters, which can be particularly beneficial for detecting mislabeled data, understanding dataset difficulty and informativeness. Empirically, the authors validate LSI across a range of datasets and demonstrate that it transfers well from smaller “probe” models to larger architectures.

**Strengths:**

* **Originality and Quality**
The approach is novel because it uses a post-hoc Bayesian approximation to evaluate sample informativeness in standard neural networks. The authors include experiment results on multiple datasets, from CIFAR-10 to IMDb, showing that LSI can detect mislabeled samples and assess informativeness across classes.

* **Clarity and Significance**
The paper is well-organized and easy to follow. As claimed in the paper, LSI is beneficial for identifying potentially mislabeled data, as well as evaluating dataset difficulty and informativeness. The proposed method could have a substantial impact, particularly for data-centric AI and applications where dataset quality is key.

**Weaknesses:**

**1. Computational Cost**

Calculating LSI requires Hessians, which may be expensive on large datasets. While the probe model helps, a more detailed breakdown of computation time, evaluation quality, and scalability would be helpful for readers to understand the effectiveness of LSI.

**2. Limitations of Probe Model**

While using a smaller probe model makes computation faster, it could introduce discrepancies in sample rankings when applied to larger models. Could the authors provide more insight into how well LSI rankings from the probe model align with those from larger models?

**3. Understanding the Effectiveness of LSI**

During the investigation of low/high LSI samples, is it possible to have statistical measurements about the comparisons/differences between high and low LSI samples. It is not clear if the observation of the difference among a few samples can generalized to the larger scale. Besides, there are no baseline performances reported in this paper. (i.e., for the detection of mislabeled samples, I believe there are certain methods already working on it.)

**4. Experiments on Real-world Mislabeled Samples**

In the section ```Focusing on Mislabeled Samples```, it would be more beneficial if authors could test the effectiveness of LSI on real-world noise (CIFAR-10N [R1], or CIFAR-10H [R2]), instead of synthetic ones (random label flips).

**References:**

R1: Learning with Noisy Labels Revisited: A Study Using Real-World Human Annotations. ICLR 2022.

R2: Human uncertainty makes classification more robust. ICCV 2019.

**Questions:**

Pleaser refer to the weakness section.

---

> ### Author Response · Authors · 2024-11-19
> **Author Comment on the Weaknesses described by Reviewer 6Bs2**
>
> Thank you for reviewing our work and pointing out strengths and weaknesses.
>
> ---
>
> > Computational Cost
>
> Thank you for the comment, you are raising a valid point. We approximate the Hessian with its diagonal which is far more efficient to compute. However, our method is not constrained to a specific type of Hessian approximation, indicating that future advances in Hessian approximation will also benefit the computation of LSI. An experiment showing that the diagonal approximation correlates well with the full Hessian for a model small enough for the full Hessian to be tractable is already part of the manuscript (Appendix  B). Moreover, we also tested the correlation between the diagonal approximation and a Kronecker-factored Hessian approximation (KFAC), which is efficient enough to be used even in very large models (see e.g. Grosse et al, Studying Large Language Model Generalization with Influence Functions, 2023). In all cases, we found that the diagonal Hessian correlates excellently. To address your comment specifically, we will also add the following table to the manuscript in Appendix B:
>
> | **Hessian Approximation** | **Time Complexity**       | **Inversion Time Complexity**    | **Space Complexity**           | **Approximation Quality**                                                                 |
> |---------------------------|---------------------------|-----------------------------------|---------------------------------|------------------------------------------------------------------------------------------|
> | **Diagonal Hessian**      | O(n)                     | O(n)                             | O(n)                           | Strong correlation with full Hessian for tested cases (Spearman’s R of 0.93)            |
> | **KFAC Approximation**    | O(n²) (with sparsity)    | O(n²) (blockwise)                | O(n²) (with sparsity)          | Strong correlation with full Hessian for tested cases (Spearman’s R of 0.93)            |
> | **Full Hessian**          | O(n³)                    | O(n³)                            | O(n²)                          | Exact representation but computationally infeasible for large models                    |
>
> Moreover, we show an excellent correlation in approximation quality of the probe model to training a full model in Appendix C. The speedup of the LOO retraining through employing the probe model depends on the size of the initial model and can thus not directly be reported as an attribute of the method, but was on the order of three magnitudes in our experimental settings, as stated in the original manuscript in the experimental section in line 252.
>
> ---
>
> > Limitations of Probe Model
>
> Thank you for the comment. We provide data supporting this claim in Appendix C of the original manuscript. Concretely, we compute Spearman’s Rank correlation coefficient between LSI scores computed using the probe model vs. LSI scores computed on a full model (CNN). Here, we find an excellent correlation, suggesting that the probe model accurately captures the LSI computed on the full model.
>
> ---
>
> > Understanding the Effectiveness of LSI
>
> Thank you for your comment. We are unsure about the meaning of “statistical measurements about the comparisons/differences between high and low LSI samples.” If our response below does not fully address your concern, we would appreciate further clarification so we can provide a more detailed answer.
> In the paper, we provide extensions of our LSI computed on individual samples to subsets of data. Section 4.3 shows that the visual correctness of our LSI extends to subsets of data, demonstrating the consistency of our findings. Larger-scale experiments using Vision and Language Transformers (CLIP) are presented in Section 4.2, showing the scalability of our approach.
> Additionally, we compare LSI to other metrics, including Pointwise Sliced Mutual Information and Smooth Unique Information in Appendix A. These results indicate that LSI is a better approximation of the mutual information than any of these methods. Moreover, other than these metrics, LSI remains computationally tractable.
> On the topic of mislabelled data detection, we would like to point out that this was not the primary focus of our paper, which is focused on data attribution. We thus used LSI as an informativeness measure to assess the impact of mislabeled data. Our experiments show that the unique sample information as measured by LSI increases with label flipping for most samples.

---

> ### Author Response · Authors · 2024-11-19
> **Author Comment on the Weaknesses described by Reviewer 6Bs2 continued**
>
> > Experiments on Real-world Mislabeled Samples
>
> Thank you for this recommendation. We have conducted an additional experiment of detecting mislabeled data on Cifar-10N, and found the following. Similar to label flipping, samples with human labeling errors, on average, have higher LSI as they carry more unique information. However, the difference to correctly labeled samples is not as big as with label flipping. This follows the intuition of unique information, as a distant brown animal (correct label: deer, given wrongfull label: dog) does not carry as much unique information, as there are also other images of distant brown animals labeled as dogs, as if it would have been labeled as an airplane. The corresponding figure will be included in the manuscript appendix. Note that Cifar-10H is unsuitable for this purpose as it solely provides mislabeled targets for the test split of the dataset and can, therefore not be investigated.

---

> ### Author Response · Authors · 2024-11-25
> **Feedback on Rebuttal Response**
>
> Dear Reviewer 6Bs2,
> Thank you for your review. At this point, our progress depends on your response. We would greatly appreciate it if you could share your thoughts on whether our reply to your review addresses some of your concerns or provide guidance on how we might resolve any remaining issues before the end of the rebuttal period.

---

> > ### Comment · Reviewer_6Bs2 · 2024-11-26
> > **Thanks for the author response**
> >
> > Thanks authors for the detailed responses. Most of my concerns are addressed and I am raising my score from 5 to 6.
> > In the revision, it would be much better if the authors could include a section either in the appendix or the main body, to discuss the observations on the real-world data (Cifar-10N). The observations/discussions would be beneficial for the researcher in the detection of mis-labeled samples, or understanding the gap between synthetic and real-world mislabeled samples.

---

> > > ### Author Response · Authors · 2024-11-26
> > > **Revised Version of the Submission**
> > >
> > > Thank you for raising the score and the additional feedback, which we have now taken into account.
> > >
> > > At the time of our comment we had not yet uploaded a revised version of the document. We agree, that a discussion on how LSI changes with humanly annotated and thus mislabeled data is interesting and can benefit researchers.
> > > We have now uploaded a revised version with the investigation of LSI computed on CIFAR-10N in **Appendix I**, which includes the corresponding figure and a short discussion of the results.
> > >
> > > *(We chose to append it here to not mess with references on appendices of the original manuscript we made in the rebuttel and will reposition it more to the front of the apendix in the camera ready version of the work.)*
> > >
> > > Moreover, the revised Version now also includes a comparison of LSI with TRAK, Park et al. 2023 (Appendix H) and is overall smaller in size, to aid with readability.

---

### Official Review · Reviewer_j9mc · 2024-11-12

**Soundness:** 2
**Presentation:** 3
**Contribution:** 2
**Rating:** 6
**Confidence:** 3

**Summary:**

This paper proposes a measure named Laplace Sample Information (LSI) to estimate the informativeness of individual samples in a dataset for deep learning methods. Specifically, LSI leverages a Bayesian approximation to the weight posterior and the KL divergence to measure the change in the parameter distribution induced by a leave-one-out sample from the dataset. Experimental results show that LSI is effective in ordering the data w.r.t. typicality, detecting mislabeled samples, measuring class-wise informativeness, and assessing dataset difficulty.

**Strengths:**

1. Overall, this paper is well-written and easy to follow.
2. This paper considers the important task of estimating the sample information.
3. Experimental results illustrate the effectiveness of the proposed measure LSI.

**Weaknesses:**

1. The novelty of the proposed method might be limited as the Laplacian approximation is widely used in the Bayesian machine learning community.
2. Theoretically, formal results are lacking to support the effectiveness of the proposed method.
3. Large experiments involving large datasets and models are lacking, which results in that this work might have little influence in practice.

Minor issues:
1. Typos exist in Lines 194-195: shownAppendix.

**Questions:**

None

---

> ### Author Response · Authors · 2024-11-19
> **Author Comment on the Weaknesses described by Reviewer j9mc**
>
> Thank you for reviewing our work and pointing out strengths and weaknesses.
>
> ---
>
> > The novelty of the proposed method might be limited as the Laplacian approximation is widely used in the Bayesian machine learning community.
>
> Thank you for your comment. Your criticism here seems to be centered on the novelty of the Laplace approximation itself rather than on how we have applied it in our work. The novelty of our work is using the Laplace approximation as a foundational tool for determining sample informativeness. This reliance on a proven method is a strength, ensuring our approach is built on a solid theoretical basis. To the best of our knowledge, the Laplace approximation has not been employed in the context of data attribution. We hope this clarifies the originality and value of our proposed method.
>
> ---
>
> > Theoretically, formal results are lacking to support the effectiveness of the proposed method.
>
> Thank you for your feedback. The assertion that theoretical support is lacking is not correct. While we do not provide a standalone proof of the KL divergence's properties, this is unnecessary, as they are well-established in the cited literature. Moreover, the LSI is based on sample information (SI) as proposed by [1]. It thus formally approximates the pointwise conditional mutual information of the samples and the weights of the neural network. LSI provides a computationally tractable implementation of the theoretically well-motivated SI which can be used for data attribution, at which we succeed. For a detailed formal derivation of sample information from the pointwise conditional mutual information, we refer you to [1], as referenced in line 149 of the original manuscript. The derivation of LSI from SI is provided in the manuscript in section 3.
>
> ---
>
> > Large experiments involving large datasets and models are lacking, which results in that this work might have little influence in practice.
>
> We appreciate your feedback regarding the scale of the experiments. We acknowledge that our work does not yet demonstrate applicability at the scale of very large datasets, such as those on the scale of ImageNet. However, we believe it is important to view this limitation within the broader context of data attribution research, particularly for methods that offer rigorous theoretical guarantees.
> Large-scale experiments are a challenge for most data attribution methods that aim to provide formal guarantees due to the inherent trade-offs between computational cost, tractability, and accuracy ([1] is solely capable of acting on subsets of a few thousand samples, [2] provides results solely on MNIST and a 900 sample subset of ImageNet, [3] requires training of hundreds of models and results are provided solely on CIFAR-10 and FMoW, [4] reduces the data attribution tasks to logistic regression, [5] required the training of hundreds of models from scratch with the number of models increasing with the dataset size). Our work addresses these trade-offs by making significant progress toward practical applicability, even if scaling to the largest datasets remains beyond the current scope. Given that our method is parallelizable, our method is applicable to arbitrary datasets and models without encountering computability issues, contrary to methods which suffer from memory scaling limitations. Thus, the primary constraint limiting our ability to conduct experiments at larger scales is computational resources rather than any intrinsic limitation of our method.
>
> ---
>
> > Typos exist in Lines 194-195: shownAppendix.
>
> Thank you for spotting the typo, we will fix it.
>
> ---
>
> [1] Harutyunyan, Hrayr, et al. "Estimating informativeness of samples with smooth unique information." (2021).
>
> [2] Koh, Pang Wei, and Percy Liang. "Understanding black-box predictions via influence functions." International conference on machine learning. PMLR, 2017.
>
> [3] Ilyas, Andrew, et al. "Datamodels: Predicting predictions from training data." (2022).
>
> [4] Park, Sung Min, et al. "Trak: Attributing model behavior at scale." (2023).
>
> [5] Vitaly Feldman and Chiyuan Zhang. “What Neural Networks Memorize and Why: Discovering the Long Tail via Influence Estimation”. In: Advances in Neural Information Processing Systems (NeurIPS). Vol. 33. 2020, pp. 2881–2891

---

> > ### Comment · Reviewer_j9mc · 2024-11-29
> >
> > Thanks for the authors' response. They have addressed most of my concerns, and I would like to increase the score to 6.

---

> ### Author Response · Authors · 2024-11-25
> **Feedback on Rebuttal Response**
>
> Dear Reviewer j9mc,
>
> Thank you for your valuable feedback. At this stage, our progress hinges on your response. We would greatly appreciate it if you could share your thoughts on whether our reply to your review resolves some of your concerns or provide guidance on how we might address any remaining issues before the rebuttal period concludes.

---

### Author Response · Authors · 2024-11-23

Dear Reviewers,
we hope that we have resolved all questions and concerns. If there are any further comments or suggestions, we are happy to engage in discussion and address them before the rebuttal period ends.

Best regards,
The Authors

---

### Meta-Review · Area_Chair_3K2M · 2024-12-21

**Metareview:**

This work introduces Laplace Sample Information (LSI), an information-theoretic measure of sample informativeness based on Bayesian approximations and KL divergence to assess a sample's impact on a model's parameter distribution. LSI effectively ranks data by typicality, detects mislabeled samples, evaluates class-wise informativeness, and measures dataset difficulty, demonstrated across image and text data in both supervised and unsupervised settings. The method is computationally efficient and transferable to large-model training.

This paper is a borderline paper. However, I think the issues raised by the reviewers can be addressed in the final version. I would encourage the authors to address the issues raised including comparison to existing approaches and moving some of the baseline comparisons to the main version from the appendix.

**Additional Comments On Reviewer Discussion:**

This paper is a borderline paper. However, I think the issues raised by the reviewers can be addressed in the final version. I would encourage the authors to address the issues raised including comparison to existing approaches and moving some of the baseline comparisons to the main version from the appendix.

---

### Decision · Program_Chairs · 2025-01-22

Accept (Poster)